# *Saccharomyces cerevisiae TAD1* Mutant Strain As Potential New Antimicrobial Agent: Studies on Its Antibacterial Activity and Mechanism of Action

**DOI:** 10.3390/microorganisms13122848

**Published:** 2025-12-15

**Authors:** Yu Zhang, Mengkun Li, Shulei Ji, Liu Cong, Shanshan Mao, Jinyue Wang, Xiao Li, Tao Zhu, Zuobin Zhu, Ying Li

**Affiliations:** 1School of Medical Technology, Xuzhou Medical University, Xuzhou 221004, China; 2Department of Genetics, Xuzhou Medical University, Xuzhou 221004, China

**Keywords:** *TAD1*, *Saccharomyces cerevisiae*, *Escherichia coli*, *Staphylococcus aureus*, *Salmonella typhi*, antimicrobial activity, organic acid

## Abstract

Human infections caused by pathogenic bacteria remain a major global health concern. Among them, *Staphylococcus aureus*, *Escherichia coli*, *Klebsiella pneumoniae*, and *Salmonella typhi* are particularly prevalent and associated with significant morbidity and mortality. While antibiotics have long been the cornerstone of bacterial infection treatment, the widespread and often inappropriate use of these drugs has led to the emergence of multidrug-resistant (MDR) strains. This escalating resistance crisis underscores the urgent need for alternative therapeutic strategies. Amid the escalating global antimicrobial-resistance crisis, a genome-wide screen of 1800 *Saccharomyces cerevisiae* knockouts identified a *TAD1*-deficient mutant whose cell-free supernatant (CFS) rapidly eradicates multidrug-resistant *E. coli*, *S. aureus*, *K. pneumoniae*, and *S. typhi* in vitro. CFS disrupts pathogenic biofilms, downregulates biofilm-associated genes, and exerts bactericidal activity by triggering intracellular reactive oxygen species (ROS) accumulation and compromising envelope integrity. Probiotic profiling revealed robust tolerance to an acidic pH and physiological bile, high auto-aggregation, and efficient co-aggregation with target pathogens. In both *Galleria mellonella* and murine infectious models, administration of CFS or live yeast significantly increased survival, attenuated intestinal histopathology, and reduced inflammatory infiltration. These data establish the *TAD1*-knockout strain and its secreted metabolites as dual-function antimicrobial-probiotic entities, offering a sustainable therapeutic alternative to conventional antibiotics against multidrug-resistant bacterial infections.

## 1. Introduction

Antibiotic resistance has become a global public health crisis, with the World Health Organization (WHO) predicting that drug-resistant infections could kill 10 million people a year by 2050 [1]. The limitations of traditional antibiotics have prompted the scientific community to turn to the development of natural antimicrobials. *Saccharomyces cerevisiae* is a class of safe and easily cultured eukaryotic microorganisms whose metabolites (such as CFS, secreted proteins, and polysaccharides) exhibit broad-spectrum antibacterial activity while having probiotic properties (such as regulating host immunity and intestinal flora balance) [2,3] and are thus seen as a potential new solution to fight bacterial infections. For example, *Saccharomyces boulardii* CNCM I-745 has been successfully used to treat diarrhea and intestinal infections by mechanisms involving the secretion of antitoxin proteins and the regulation of host immune responses [4,5].

*S. cerevisiae* shares a close relationship with *S. boulardii* and possesses potential for development as a probiotic. In recent years, research on the antibacterial mechanism and application of *S. cerevisiae* has made remarkable progress. The antimicrobial activity of *S. cerevisiae* is mainly due to its metabolites, including organic acids (such as lactic acid [6]), secreted proteins (such as lysozyme [7]), polyphenols (such as resveratrol), and polysaccharides (such as β-glucan [8]). These components can inhibit the growth of pathogenic bacteria by disrupting the integrity of bacterial cell membranes, inhibiting biofilm formation, and interfering with quorum-sensing systems (such as downregulating the virulence gene *lasR*). For example, it was found that *S. cerevisiae* CFS had a minimum inhibitory concentration (MIC) as low as 8% against methicillin-resistant *Staphylococcus aureus* (MRSA) and was able to inhibit biofilm formation and downregulate the expression of related genes [9]. In addition, *S. cerevisiae* can enhance host immunity by activating macrophages and enhancing T cell responses [10,11], and its probiotic properties further expand its potential application in the treatment of infectious diseases. Genetic engineering technologies have also significantly improved the antimicrobial properties of *S. cerevisiae*. For example, the high-tolerance strain, XP2, screened by ARTP mutagenesis, increased its biomass in corn stalk hydrolysate by 19.68%, enhancing its adaptability to environmental stress [12]. CRISPR-Cas9 technology has been used to optimize yeast strains, for example, by overexpressing β-glucan synthase to enhance the production of antimicrobial components [13].

Despite these remarkable achievements, several challenges remain in elucidating the antimicrobial mechanisms of *S. cerevisiae*. The synergistic antibacterial effects of various bioactive components (such as organic acids and polyphenols) in CFS have not been systematically analyzed. Most existing studies are limited to in vitro experiments, with insufficient validation in animal models and clinical settings. The adaptability of *S. cerevisiae* to complex infection environments still needs to be optimized, and the antimicrobial activity varies significantly among different strains. In preliminary work, we systematically evaluated the CFS antibacterial activity of 1800 *S. cerevisiae* single-gene-knockout strains [The Saccharomyces cerevisiae gene-knockout collection (YKOC)] using *Escherichia coli (mcr-1)* 12-2 as an indicator strain, with the CFS of parental BY4743 serving as a negative control. This screening yielded 303 mutants exhibiting significant antibacterial activity. Enrichment analysis revealed that the *TAD1* gene resides at the core module of the differential gene interaction network. The Saccharomyces Genome Database (https://www.yeastgenome.org accessed 14 October 2025.) annotates *TAD1* as an RNA-specific adenosine deaminase (Tad1p/scADAT1), responsible for catalyzing the post-transcriptional modification of tRNAAla at position 37 (adenosine → inosine, A37I), representing an upstream event in tRNA functional regulation. Given that Tad1p-mediated modification occurs at the inception of protein synthesis quality control, its absence may indirectly enhance the synthesis and export of yeast secretory antimicrobial molecules by reshaping the translational landscape, activating non-canonical stress pathways, or reprogramming secondary metabolism. Consequently, a causal link between *TAD1* deficiency and the antibacterial phenotype warrants further functional validation and mechanistic elucidation. Given the current paucity of research into the function of the *TAD1* gene, this study aims to investigate the antimicrobial activity and probiotic potential of *S. cerevisiae TAD1-KO* in vitro; to elucidate the synergistic antimicrobial mechanism of multiple components in its CFS using metabolomics to identify key bioactive molecules; and to validate the antibacterial efficacy of *S. cerevisiae TAD1-KO* and its metabolites in vivo by using a mouse infection model. Through these efforts, we sought to further explore the probiotic potential of *S. cerevisiae TAD1-KO* and provide novel insights for alternative therapeutic strategies against bacterial infections.

## 2. Materials and Methods

### 2.1. Strains and Growth Conditions

YKOC was obtained from Invitrogen (Carlsbad, CA, USA) in 2014 and maintained at −80 °C in the YPD medium with 20% (*v*/*v*) glycerol. *S. boulardii* (import registration No. S5288500) was purchased from Biocodex (Gentilly, France). Genotype of BY4743: MATa/α, his3Δ1/his3Δ1, leu2Δ0/leu2Δ0, LYS2/lys2Δ0, met15Δ0/MET15, ura3Δ0/ura3Δ0; Genotype of YGL243W (*TAD1*): kanMX4. These were obtained from the EUROSCARF collection (Scientific Research and Development GmbH, Oberursel, Germany). Clinical isolates of *E. coli*, *S. aureus*, *P. aeruginosa*, *K. pneumoniae*, *A. baumannii*, and *S. typhi* were provided by the Affiliated Hospital of Xuzhou Medical University and routinely cultured in LB broth at 37 °C.

### 2.2. Preparation of CFS from S. cerevisiae

Single-gene-deletion mutants and the parental BY4743 strain, stored at −80 °C, were revived in YPD broth and incubated (200 rpm, 30 °C, 16 h). Cultures were centrifuged (4000× *g*, 4 °C, 12 min), and the supernatants filtered through 0.22 µm syringe filters. CFS were adjusted to the recorded pH values, aliquoted, and stored at −80 °C; BY4743 CFS served as the control.

### 2.3. Establishing Yeast Growth Curves

*S. cerevisiae TAD1-KO*, BY4743, and *S. boulardii* were cultivated in YPD medium at 30 °C, 200 rpm for 24 h. Every 2 h, 100 µL aliquots were withdrawn, and the OD_600_ was recorded on a multi-mode microplate reader to construct growth curves.

### 2.4. Growth Curve

Log-phase pathogens (pre-cultured overnight) were adjusted to 1 × 10^6^ CFU mL^−1^ in LB medium. Bacterial suspensions were mixed 9:1 (*v*/*v*) with either *S. cerevisiae TAD1-KO* CFS or BY4743 CFS (test groups), YPD medium (vehicle control), or sterile water (blank control). An initial OD_600_ was recorded; cultures were then incubated at 30 °C, 200 rpm, and the OD_600_ was monitored every 2 h to construct growth curves.

### 2.5. Inhibitory Activity of S. cerevisiae TAD1-KO CFS Against Different Bacteria

Nine indicator strains—*E. coli* ATCC 25922, *E. coli* (*mcr-1*), *S. aureus* ATCC 29213, *S. aureus* ATCC 25923, MRSA, *P. aeruginosa* 1554, *K. pneumoniae* 2118, *A. baumannii* 21-1, and *S. typhi* SL1344—were screened for susceptibility. In a 96-well plate, 90 µL of CFS was combined with 10 µL bacterial suspension (1 × 10^8^ CFU mL^−1^); CFS from BY4743 and *S. boulardii* served as the controls. After incubation at 37 °C, 90 rpm for 18–24 h, the OD_600_ was read. The growth rate was calculated as (OD_sample_/OD_control_), and inhibition was defined as ≥80% reduction (1 – growth rate ≥ 0.8).

### 2.6. Biofilm Inhibition Assay

Biofilm inhibition by *S. cerevisiae TAD1-KO* CFS was quantified with crystal violet (CV) staining. *E. coli* ATCC 25922, *E. coli* (*mcr-1*), *S. aureus* ATCC 29213, MRSA, and *S. typhi* SL1344 were grown to 1 × 10^6^ CFU mL^−1^ in 96-well plates containing 25%, 50%, or 90% (*v*/*v*) *TAD1-KO* CFS; the YPD medium served as a control. After 18–24 h at 37 °C, planktonic cells were aspirated and the wells were gently washed twice with 150 µL of sterile water. Adherent biofilms were fixed (37 °C, 30 min), stained with 0.1% CV (150 µL, 30 min), rinsed twice, and destained with 150 µL of 30% methanol/10% acetic acid. The OD_600_ was recorded, and the inhibition was calculated as:Biofilm inhibition rate (%) = (1 − OD_treatment/_OD_control_) × 100

### 2.7. Expression of E. coli Biofilm-Related Genes

Overnight cultures of *E. coli* were diluted to 1 × 10^8^ CFU mL^−1^ in LB and further adjusted to 1 × 10^7^ CFU mL^−1^ in either 50% (*v*/*v*) *TAD1-KO* CFS (test group) or LB (control group), then incubated for 12 h at 37 °C. Cells were pelleted (10,000× *g*, 4 °C, 5 min). After treatment, total RNA was extracted by TRIzol (TIANGEN Biotechnology, Beijing, China), and complementary DNA (cDNA) was synthesized by the cDNA synthesis kit. Then, real-time fluorescent quantitative PCR (RT-PCR) with a specific primer reverse-transcription polymerase chain reaction was used to evaluate the expression of biofilm-related genes (Appendix A). The results were calculated by 2^(−ΔΔCt)^.

### 2.8. Adhesion Ability to the Glass Surface

Overnight cultures of *E. coli*, *S. aureus*, and *S. typhi* were diluted in LB to 1 × 10^6^ CFU mL^−1^ and mixed 1:1 with either 50% *TAD1-KO* CFS (test group) or YPD (control group). Aliquots (4 mL) were transferred to glass tubes inclined at 30° and incubated for 24 h at 37 °C. Following incubation, planktonic cells were carefully removed and retained. Adherent cells were gently rinsed twice with PBS, detached, and resuspended in fresh PBS to quantify the biofilm biomass (OD_adhered cell_). The retained planktonic fraction was then combined with the detached cells, washed once, and resuspended in PBS to measure total cell density (OD_cell mixture_). We used the following equation to calculate the adhesion capacity (%):Adhesion ability (%) = (OD_adhered cell_/OD_cell mixture_) × 100

### 2.9. Hydrophobicity Assay

Overnight cultures of *E. coli*, *S. aureus*, and *S. typhi* were diluted in LB to 1 × 10^6^ CFU mL^−1^ and incubated for 24 h at 37 °C with either 50% (*v*/*v*) *TAD1-KO* CFS (test group) or an equal volume of YPD (control group). Cells were harvested (17,709× *g*, 4 °C, 5 min), washed twice with PBS, and resuspended to OD_600_ = 0.50 ± 0.05 (OD_initial value_). For cell-surface hydrophobicity, 1.2 mL of this suspension was vortexed with 0.3 mL of *n*-octane for 3 min; phases were allowed to separate for 15 min, and the OD_600_ of the aqueous phase (OD_treatment_) was recorded. We used the following equation to calculate the hydrophobicity (%):Hydrophobicity (%) = (1 − OD_treatment_/OD_initial value_) × 100

### 2.10. Exopolysaccharides (EPS) Production

The procedure is similar to the one described in Section 2.9. After incubation, cultures were centrifuged (8000× *g*, 4 °C, 10 min), washed twice with PBS, and resuspended in 1 mL of 0.9% saline. An equal volume of 5% phenol was added, followed by 5% sulfuric acid, and the mixture was incubated in the dark for 1 h. Optical density was then measured at 490 nm. We used the following equation to calculate the EPS quantification (%):EPS quantification (%) = (OD_treatment_/OD_control_) × 100

### 2.11. Kinetics of E. coli Killing by CFS

An overnight culture of *E. coli* was diluted to 1 × 10^8^ CFU mL^−1^ in LB and further adjusted to 1 × 10^7^ CFU mL^−1^ in either 90% BY4743 CFS, 90% *TAD1-KO* CFS (test groups), or LB (control). Cultures were incubated at 37 °C, 200 rpm. At predetermined intervals, aliquots were serially diluted, plated in triplicate on LB agar, and incubated at 37 °C for 12 h. According to the number of monoclonal colonies growing on the plate, the number of viable bacteria in the original bacterial solution was calculated, and the time–kill curve was drawn.

### 2.12. Transmission Electron Microscopy (TEM) Analysis

*E. coli* and *S. aureus* were cultured overnight to the logarithmic phase. The pathogenic bacteria suspension was then co-cultured with different amounts of *S. cerevisiae TAD1-KO* CFS at 37 °C, 200 rpm for 8–10 h. The group without CFS was used as the control group. The bacterial precipitate was collected by centrifugation, and the medium was discarded and fixed with electron microscope fixative for 30 min at room temperature and protected from light, and then transferred to 4 °C for storage.

### 2.13. Effect of S. cerevisiae TAD1-KO CFS on Cell Wall Integrity of E. coli

The overnight culture of *E. coli* was adjusted to 1 × 10^8^ CFU mL^−1^ and further diluted to 1 × 10^7^ CFU mL^−1^ in either 90% or 50% (*v*/*v*) *TAD1-KO* CFS (test groups) or LB (control). Cultures were incubated at 37 °C, 180 rpm, and samples were withdrawn at 0, 2, 4, 6, and 8 h. Cells were centrifuged (10,000× *g*, 5 min), and the absorbance of the bacterial solution at 520 nm was measured using the AKP Assay kit (Beyotime, Shanghai, China). Extracellular AKP activity was calculated according to the manufacturer’s recommendations.

### 2.14. Determination of Extracellular Membrane Permeability

The overnight culture of *E. coli* was adjusted to 1 × 10^8^ CFU mL^−1^ and diluted to 1 × 10^7^ CFU mL^−1^ in either 50% (*v*/*v*) *TAD1-KO* CFS (test group), LB (negative control), or 1 mg mL^−1^ EDTA·2Na (positive control). After 6 h at 37 °C, 180 rpm, cells were centrifuged (10,000× *g*, 4 °C, 10 min), washed, and resuspended in PBS to OD_600_ = 0.50 ± 0.05. For the N-Phenyl-1-naphthylamine (NPN) uptake assay, 180 µL of each suspension was transferred to a black 96-well plate, mixed with 20 µL NPN (final 100 µM), and incubated for 1 h at room temperature in the dark. Fluorescence was recorded at λₑₓ 350 nm/λₑₘ 450 nm.

### 2.15. Determination of Intracellular Membrane Permeability

Log-phase *E. coli* (overnight culture) was centrifuged (12,000× *g*, 4 °C, 10 min) and resuspended in PBS to OD_600_ = 0.20 ± 0.02. 170 µL aliquot of this suspension was dispensed into each well of a 96-well plate, followed by 10 µL 30 mM O-Nitrophenyl-β-d-galactopyranoside (ONPG) and 20 µL of either *TAD1-KO* CFS (test group), PBS (negative control), or 1.25 mg mL^−1^ polymyxin B (positive control). ONPG hydrolysis was monitored every 20 min by measuring the absorbance of the resulting O-nitrophenol at 415 nm.

### 2.16. Reactive Oxygen Species (ROS) Relative Fluorescence Intensity Determination

The overnight culture of *E. coli* was adjusted to 1 × 10^8^ CFU mL^−1^ and diluted to 1 × 10^7^ CFU mL^−1^ in either 90% or 50% (*v*/*v*) *TAD1-KO* CFS (test group) or LB (control), followed by a 6 h incubation at 37 °C. The OD_600_ was recorded (100 µL per well, 96-well plate) before the cells were centrifuged (10,000× *g*, 4 °C, 5 min). Pellets were loaded with DCFH-DA (10 µM, 37 °C, 30 min, dark) according to the ROS Assay Kit (Beyotime, Shanghai), washed once with PBS, and resuspended in fresh buffer. ROS-derived fluorescence was read at λₑₓ488 nm/λₑₘ525 nm. We used the following equation to calculate the relative fluorescence intensity:Relative fluorescence intensity = fluorescence intensity/OD_600_

### 2.17. Bacterial Intracellular ROS Inhibition Assay

The overnight culture of *E. coli* was adjusted to 1 × 10^8^ CFU mL^−1^, supplemented with Vitamin C (VC) or thiourea (Tu) at a final concentration of 1 mmol L^−1^ or 2 mmol L^−1^, and further diluted to 1 × 10^7^ CFU mL^−1^ in the following media: (i) 90% *TAD1-KO* CFS, (ii) 90% *TAD1-KO* CFS + VC, (iii) 90% *TAD1-KO* CFS + Tu, and (iv) 90% *TAD1-KO* CFS + VC + Tu. Cultures were incubated at 37 °C, 200 rpm. At predetermined intervals, aliquots were serially diluted, plated in triplicate on LB agar, and incubated at 37 °C for 12 h. Viable counts were enumerated from single colonies to compare the bactericidal impact of ROS scavengers on *TAD1-KO* CFS activity.

### 2.18. Probiotic Potential of S. cerevisiae TAD1-KO

(1)Tolerance of *S. cerevisiae TAD1-KO* to artificially simulated gastroenteric fluid:

Gastric tolerance: pepsin (3 mg mL^−1^) in PBS, pH 3.0; intestinal tolerance: trypsin (1 mg mL^−1^) plus 0.3% bile salts in PBS, pH 8.0. Both solutions were sterile-filtered (0.22 µm). *S. cerevisiae TAD1-KO* and *S. boulardii* (positive control) were grown for 16 h in YPD (30 °C, 200 rpm), centrifuged (4000× *g*, 4 °C, 12 min), washed twice with PBS, and resuspended in the respective fluids. Viable counts were calculated by the dilution coated plate method after exposure to gastric fluid for 1 h and 3 h, or intestinal fluid for 2 h and 4 h (37 °C water bath).

(2)Auto-aggregation Ability Assay: Auto-aggregation of *S. cerevisiae TAD1-KO* and its co-aggregation with *E. coli*, *S. aureus*, and *S. typhi* were assessed in parallel, with BY4743 and *S. boulardii* serving as the reference and positive control, respectively. Yeast strains grown overnight were harvested, washed twice with PBS, and resuspended in the same buffer to an OD_600_ of ≈0.5 (A_0_). Suspensions were left undisturbed at room temperature; after 5 h; the OD_600_ of the upper phase was recorded (Aₜ). Auto-aggregation (%) was calculated using the following equation:

Auto-aggregation (%) = [1 − (A_t_/A_0_)] × 100

(3)Co-aggregation Ability Assay: Overnight cultures of *E. coli*, *S. aureus*, and *S. typhi* were harvested (8000× *g*, 4 °C, 10 min), washed twice, and resuspended in PBS to an OD_600_ ≈ of 0.5 (A1). Yeast suspensions (*S. cerevisiae TAD1-KO*, BY4743, or *S. boulardii*) were prepared identically (A2). Subsequently, 2 mL of each yeast suspension was mixed with 2 mL of the corresponding bacterial suspension, incubated statically at 37 °C for 5 h, and the OD_600_ of the upper phase was recorded (A3). Co-aggregation (%) was calculated using the following equation:

Co-aggregation (%) = [(A1 + A2)/2 − A3)/(A1 + A2)/2] × 100

### 2.19. Killing Model of the Moth Galleria mellonella

*G. mellonella* was used to evaluate the protective capacity of *S. cerevisiae TAD1-KO* against *E. coli* and *S. aureus*. Yeast cells were harvested, washed three times with PBS, and resuspended in the same buffer to obtain the “live” suspension. A 5 mL aliquot was autoclaved (121 °C, 20 min), centrifuged (8000× *g*, 10 min), washed three times, and resuspended in PBS to generate the “heat-killed” preparation. The *G. mellonella* were randomly grouped into 8 groups: 1 blank control group (PBS buffer group); 1 control group (single pathogen group); 3 toxicity test groups (CFS group, Alive group, Heat-killed group); and 3 treatment groups (CFS + pathogen group, Alive + pathogen group, Heat-killed + pathogen group). The survival rate of the *G. mellonella* for 5 days in each group was recorded after dark culture at 37 °C.

### 2.20. Mouse Model of Enteritis

(1)The enteritis model of mice infected by *S. typhi* SL1344 was established. Specific-pathogen-free grade male C57BL/6J mice, each at 6–8 weeks of age, weighing 16–20 g, were purchased from Jiangsu Jicui Pharmachem Biotechnology Co. (Nanjing, China), Laboratory Animal Production License No. SCXK (Su) 2018-0008; Laboratory Animal Use License No. SYXK (Su) 2020-0048 (barrier environment). All animal experimental protocols were approved by the Animal Ethics Committee of Xuzhou Medical University.(2)SPF male C57BL/6J mice were acclimated for one week prior to experimentation. *S. cerevisiae TAD1-KO* and BY4743 were harvested, washed twice with PBS, and resuspended at 5.0 × 10^8^ CFU mL^−1^. Thirty-six mice were randomly allocated into three groups (n = 12 each) and gavaged once daily for 7 days with 0.1 mL of: *S. cerevisiae TAD1-KO* suspension, BY4743 suspension, 0.9% saline (control). Animals were identified by tail markings; their general appearance and body weight were recorded daily. Safety endpoints were evaluated by comparing the two yeast groups with the saline control. The specific procedure showed in Figure 1.(3)Each of the three treatment groups was split into two sub-cohorts (n = 6). Sub-cohort 1 (5-day infection study)**:** Mice were challenged orally with 1 × 10^7^ CFU mL^−1^ of *S. typhi* and weighed daily. On day 5, the animals were euthanised; the ileum, colon, and caecum were aseptically removed. Intestinal tissue segments were weighed and homogenized in 1 mL of 0.1% Triton X-100. Serial dilutions were plated on an LB plate supplemented with 200 µg mL^−1^ of streptomycin to enumerate *S. typhi*, and on YPD plates supplemented with 200 µg mL^−1^ of chloramphenicol to quantify *S. cerevisiae TAD1-KO* colonization. Sub-cohort 2 (survival study): After an identical challenge, mice were monitored for 10 days, and survival was recorded.(4)The spleens of the three groups of mice were weighed and photographed, and the spleen index was calculated. We used the following equation to calculate the spleen index (%): spleen index (%) = (spleen weight/body weight) × 100.(5)Three groups of mouse colon tissues were taken to measure the length and were photographed.(6)Histomorphometry: The terminal 2 cm of ileum was fixed in 4% paraformaldehyde, paraffin-embedded, and sectioned at 5 µm. Hematoxylin–eosin-stained (H&E) slides were examined under light-microscopy; images were captured for assessment of villus architecture, inflammatory infiltration, and epithelial integrity.(7)Quantitative Real-Time PCR: Total RNA was extracted from the ileal segments with TRIzol (TIANGEN, Beijing, China) and quantified on a NanoDrop-1000 spectrophotometer (Thermo Fisher, Waltham, MA, US); only samples with A260/A280 ≥ 1.9 were processed. cDNA was synthesized using a reverse-transcription kit (Sangon Biotech, Shanghai, China). Quantitative PCR was performed in a Light Cycler 480 (Roche, Basel, Switzerland) with Universal SYBR qPCR Master Mix (Biosharp, Hefei, China) and gene-specific primers (Sangon Biotech; Appendix A). Transcript levels of IL-6, IL-10, TNF-α, Occludin, Claudin-1, and ZO-1 were normalized to β-actin and expressed as 2^(−ΔΔCt)^.

### 2.21. Cell–Cell Culture Model

In order to investigate whether there is an inhibitory effect of live *S. cerevisiae TAD1-KO* and BY4743 on the growth of *E. coli*, *S. aureus*, and *S. typhi*, a cell–cell co-culture model was established. Overnight cultures of *S. cerevisiae* and pathogens were washed twice with PBS and resuspended to 1 × 10^7^ CFU mL^−1^ in YPD or LB, respectively. For co-culture, 1 mL of each yeast suspension was mixed 1:1 (*v*/*v*) with 1 mL of pathogen; pathogen mono-cultures received 1 mL of YPD, and blank controls received 1 mL of sterile water. All tubes were incubated at 37 °C, 200 rpm. After 16 h and 24 h, aliquots were spiral-plated on LB agar containing 10 mg mL^−1^ of amphotericin B to enumerate viable bacteria. Experiments were performed in three independent biological replicates.

### 2.22. Transwell Experiment

In order to verify whether the secretions of *S. cerevisiae TAD1-KO* were able to inhibit the growth of pathogenic bacteria, we used the Transwell device to test this idea. *S. cerevisiae* and pathogenic bacteria (*E. coli* and *S. aureus*) cultured overnight were diluted with YPD and LB to 1 × 10^7^ CFU mL^−1^ and 1 × 10^6^ CFU mL^−1^. Firstly, 500 μL of pathogenic bacteria were added to each of the three wells of the 12-well plate, and 200 μL of *S. cerevisiae* suspension was added directly to one of the wells as the mixed group; another well was divided into small chambers, and the same amount of yeast suspension was added to the upper chamber as the separated group; and 200 μL of YPD was added directly to the last well as the control group. The cultures were incubated at 37 °C, 90 rpm. After 16 h and 24 h, aliquots were spiral-plated on LB agar containing 10 mg mL^−1^ amphotericin B to enumerate viable bacteria. Figure 2 showed the specific device diagram.

### 2.23. Mebabolomic Analysis

*S. cerevisiae TAD1-KO* and BY4743 were cultivated in parallel (YPD, 30 °C, 200 rpm, 16 h; six biological replicates per strain). Cultures were chilled on ice, centrifuged (4000× *g*, 4 °C, 12 min), and the supernatant was immediately filtered through 0.22 µm sterile syringe filters. CFS were snap-frozen in liquid nitrogen and shipped on dry ice to Shanghai Majorbio Bio-Pharm Technology Co., Ltd. (Shanghai, China) for untargeted LC-MS/MS-based metabolomic analysis. (For specific methods, please refer to Appendix A).

### 2.24. pH Neutralization Experiment

The pH of untreated *S. cerevisiae TAD1-KO* CFS was recorded, and then aliquots were adjusted to pH 2–7 with 0.1 M HCl or 0.1 M NaOH. Treated or untreated CFS was mixed 9:1 (*v*/*v*) with *E. coli* or *S. aureus* suspensions (final 1 × 10^7^ CFU mL^−1^) and incubated 24 h at 37 °C; the OD_600_ was measured to quantify growth inhibition.

### 2.25. Statistical Analysis

All experiments were set up with three replication groups and repeated three times on three different days. Statistical analysis and mapping were performed using GraphPad Prism 10 software (US). The experimental data were statistically tested using the Student’s *t*-test or one-way ANOVA test. Statistical differences were determined according to *p*-values, * *p* < 0.05, ** *p* < 0.01, *** *p* < 0.005.

## 3. Results

### 3.1. Knockout of the TAD1 Gene in S. cerevisiae Basically Does Not Result in a Drastic Change in Proliferation Rate

Consistent with previous reports that deletion of non-essential genes can impair energy metabolism, amino-acid synthesis, or DNA replication without compromising viability [14,15], we observed that *TAD1* deletion conferred no detectable growth defect: the *S. cerevisiae TAD1-KO* and BY4743 displayed superimposable growth curves over 24 h (Figure 3A), indicating that loss of *TAD1* does not significantly perturb yeast growth under standard laboratory conditions.

### 3.2. S. cerevisiae TAD1-KO CFS Inhibits Proliferation of E. coli, S. aureus, and S. typhi

Growth-curve analysis revealed a potent, strain-specific inhibitory activity of *S. cerevisiae TAD1-KO* CFS. Exposure to 90% (*v*/*v*) *S. cerevisiae TAD1-KO* CFS markedly postponed the lag-to-log transition, lowered the maximum specific growth rate (µmax), and reduced the final stationary-phase density of *E. coli*, *S. aureus*, and *S. typhi* relative to the untreated controls (*p* < 0.01). BY4743 CFS produced a weaker suppression, albeit detectable, underscoring the contribution of the *TAD1* deletion. To exclude nutrient limitation as a confounder, sterile water and full-YPD controls were run in parallel; no significant difference in growth parameters was observed between these two extremes (Figure 3B–D), confirming that the observed inhibition is attributable to bioactive metabolites rather than carbon or nitrogen depletion.

### 3.3. S. cerevisiae TAD1-KO CFS Can Also Inhibit the Proliferation of Many Pathogenic Bacteria

Antimicrobial-spectrum profiling revealed that *S. cerevisiae TAD1-KO* CFS inhibits a broad range of Gram-negative and Gram-positive pathogens, including methicillin-resistant *S. aureus* and colistin-resistant *E. coli* (inhibition ≥ 80%; Table 1). Across all indicator strains, the activity of *S. cerevisiae TAD1-KO* CFS significantly exceeded that of both BY4743 CFS and *S. boulardii* CFS (*p* < 0.01), confirming that deletion of *TAD1* endows *S. cerevisiae* with enhanced broad-spectrum antibacterial potency.

### 3.4. S. cerevisiae TAD1-KO CFS Inhibits Biofilm Formation of Pathogenic Bacteria

The formation of biofilms is a key factor in bacterial pathogenicity [16]. Previous reports showed that the supernatant and lysate of *S. cerevisiae* cultures inhibited *S. aureus* and *P. aeruginosa* biofilm formation [17,18,19,20]. Consistent with the growth inhibition data, *S. cerevisiae TAD1-KO* CFS disrupted biofilm development in a dose-dependent manner (Figure 4A). Even at 25% (*v*/*v*), CFS reduced the CV-stainable biomass below the untreated control, while 90% CFS suppressed biofilm formation to ≤16% for all tested strains: *E. coli* ATCC 25922 (10.5%), *E. coli* mcr-1 (11.1%), *S. aureus* ATCC 29213 (9.0%), MRSA (10.8%), and *S. typhi* SL1344 (15.9%). Macroscopic CV staining of 96-well plates and light-microscopy corroborated the progressive loss of biofilm architecture with increasing CFS concentration (Figure 4B,C). The results of Figure 4D showed that 50% *S. cerevisiae TAD1-KO* CFS was able to downregulate the expression of some genes, such as *fimH* [21], *fliC* [22], *csgA* [23], and *csgD* [24]. Collectively, these findings demonstrate that *S. cerevisiae TAD1-KO* CFS effectively prevents bacterial biofilm formation.

### 3.5. S. cerevisiae TAD1-KO CFS Could Reduce Adhesion, Surface Hydrophobicity, and EPS Production of E. coli, S. aureus, and S. typhi

It has been reported that components of EPS, such as polysaccharides and proteins, may affect adhesion by altering surface charge and hydrophobicity [25]. Bacteria with high features adhere more easily, and high EPS production may enhance this feature, thus promoting adhesion [26]. There are complex interactions among them, which affect the biofilm formation, environmental adaptability, and ecological function of bacteria [27,28]. Therefore, we investigated the adhesion, hydrophobicity, and EPS production of *E. coli*, *S. aureus*, and *S. typhi* after *S. cerevisiae* CFS treatment. Compared with untreated controls, 50% (*v*/*v*) CFS reduced adhesion to glass by 53.8% (*E. coli*), 63.1% (*S. aureus*), and 47.9% (*S. typhi*) (Figure 5A; *p* < 0.05). Parallel measurements of cell-surface hydrophobicity (Figure 5B) and EPS yield (Figure 5C) showed concordant decreases: hydrophobicity dropped by 22.9%, 9.4%, and 16.0%, while EPS production fell by 26.6%, 20.6%, and 11.7%, respectively (all *p* < 0.05). Thus, *S. cerevisiae TAD1-KO* CFS simultaneously suppresses adhesion, surface hydrophobicity, and EPS biosynthesis—three key prerequisites for robust biofilm development.

### 3.6. S. cerevisiae TAD1-KO CFS Has a Certain Bactericidal Effect on E. coli

Time–kill analysis revealed a dose- and time-dependent antimicrobial action of *S. cerevisiae TAD1-KO* CFS against *E. coli* (Figure 6A). Relative to the LB control and 90% BY4743 CFS, 90% *S. cerevisiae TAD1-KO* CFS suppressed proliferation within 2 h, and the bactericidal effect could be achieved at 4 h, reaching maximal bactericidal activity at 8 h. Thus, low concentrations exert bacteriostatic effects, whereas high concentrations deliver rapid bactericidal action. TEM imaging (Figure 6B) corroborated these data: both *E. coli* and *S. aureus* exposed to 50% *S. cerevisiae TAD1-KO* CFS showed markedly reduced cell density and altered surface morphology compared to the untreated controls. The bacteria showed varying degrees of swelling, contraction, rupture, leakage of contents, and plasmolysis [29,30], confirming the disruptive impact of the secreted metabolites on bacterial integrity.

### 3.7. S. cerevisiae TAD1-KO CFS Can Damage the Integrity of E. coli’s Cell Wall and Affect the Permeability of the Cell Membrane

Cell–wall biogenesis is essential for maintaining bacterial shape and mechanical integrity, shielding the organism from osmotic and environmental insults. In Gram-negative species, the outer-membrane lipopolysaccharide (LPS) not only confers structural rigidity but also serves as a key determinant of virulence [31]. Consequently, cell–wall biosynthesis is indispensable for bacterial survival and represents a validated target for antimicrobial therapy. AKP is confined to the periplasmic space; damage to the wall or outer membrane permits its release, making extracellular AKP activity a sensitive indicator of cell–wall integrity [32]. As shown in Figure 7A, exposure to 50% (*v*/*v*) CFS elicited a time-dependent increase in extracellular AKP activity that became statistically significant at 4 h (*p* < 0.01) and continued to rise thereafter. This rapid leakage indicates that CFS compromises cell–wall integrity, allowing periplasmic enzymes to escape into the medium, and provides a mechanistic explanation for the observed antibacterial activity.

NPN is a hydrophobic fluorophore that emits only after partitioning into lipid bilayers. In intact Gram-negative cells, the outer-membrane LPS excludes NPN; disruption of this barrier permits probe penetration and fluorescence. Thus, the increment in NPN fluorescence provides a quantitative index of outer-membrane permeabilization: the higher the intensity, the greater the damage [33]. As illustrated in Figure 7B, 50% (*v*/*v*) CFS evoked a marked rise in NPN fluorescence that was statistically indistinguishable from the positive control (1 mg mL^−1^ EDTA-2Na; *p* > 0.05) and significantly exceeded the basal signal of untreated bacteria (*p* < 0.001). These data demonstrate that CFS compromises outer-membrane integrity, leading to a permeability increase comparable to that induced by a classical membrane-disrupting agent.

ONPG is a chromogenic probe used to quantify inner-membrane permeability. Under lactose induction, bacteria synthesize β-d-galactosidase, which is normally confined to the cytoplasm. When the cytoplasmic membrane is compromised, ONPG enters the cell and is hydrolyzed to galactose and the yellow chromophore o-nitrophenol (ONP) (λ_max_ = 415 nm); the rate of color development is, therefore, directly proportional to inner-membrane leakage [34]. The increase in OD_415_ mirrors the accumulation of ONP and thus reflects both β-d-galactosidase activity and inner-membrane integrity. As shown in Figure 7C, the release of ONP in the *S. cerevisiae TAD1-KO* CFS treatment group was slightly higher (*p* < 0.05) than that in the control group, indicating that CFS perturbs the cytoplasmic membrane sufficiently to allow ONPG influx and subsequent enzymatic hydrolysis.

### 3.8. ROS As a Byproduct: S. cerevisiae TAD1-KO CFS Kills Bacteria via Direct Cell Structure Damage

ROS are a class of chemically active molecules containing oxygen [35]. In biology, ROS is a key mediator of host immune defense (e.g., bactericidal effect [36]) and may also cause oxidative stress damage (e.g., DNA breakage, lipid peroxidation [37]). Intracellular ROS were quantified with the DCFH-DA fluorescence probe. As shown in Figure 7D, 90% (*v*/*v*) CFS evoked a marked increase in relative fluorescence intensity compared with the untreated control (*p* < 0.001), indicating that membrane disruption allows probe influx and reflects ROS accumulation inside the cells. To determine whether intracellular ROS accumulation is the primary driver of CFS-mediated bacteriostasis, we supplemented 90% *TAD1-KO* CFS with the ROS scavengers VC and thiourea. Plate counts after 4 h and 8 h revealed no statistically significant difference in *E. coli* viability relative to the CFS-only control (Figure 7E,F), although a numerical reduction was observed. This finding indicates that ROS are largely a secondary consequence of CFS-induced envelope disruption rather than the principal lethal agent; the dominant bactericidal effect is attributable to direct structural damage of the cell membrane and wall.

### 3.9. S. cerevisiae TAD1-KO Has Good Probiotic Properties

#### 3.9.1. *S. cerevisiae* TAD1-KO Had Good Tolerance to Artificial Gastric and Intestinal Fluids

The tolerance of gastroenteric fluid directly determines whether probiotics can survive and colonize in the harsh digestive tract environment [38]. Strains that lack tolerance are destroyed by stomach acid, bile, or digestive enzymes and are unable to perform functions such as regulating the flora, strengthening the barrier, or immune regulation [39]. Tolerance to gastric acidity and bile is a prerequisite for any candidate probiotic. Following 3 h of exposure to simulated gastric fluid (pH 3.0, pepsin 3 mg mL^−1^), *S. cerevisiae TAD1-KO* retained 86.0% viability, a value only modestly lower than that of the reference strain *S. boulardii* (Figure 8A). Similarly, after 4 h in artificial intestinal fluid (pH 8.0, trypsin 1 mg mL^−1^, 0.3% bile salts), survival remained at 80.7% (Figure 8B). These data demonstrate that *S. cerevisiae TAD1-KO* withstands the physicochemical stresses of the upper gastrointestinal tract and is competent for in vivo colonization.

#### 3.9.2. Auto-Aggregation and Co-Aggregation

The ability of auto-aggregation and co-aggregation is the key for probiotics to survive and function in the complex intestinal environment [40]. The former enhances the function of flora by enhancing colonization stability and synergistic metabolism, while the latter directly interferes with the adhesion and proliferation of pathogenic bacteria [41]. Auto-aggregation and co-aggregation are key functional traits that facilitate intestinal colonization and pathogen exclusion. *S. cerevisiae TAD1-KO* exhibited 23% higher auto-aggregation than BY4743 (Figure 8C) and formed significantly denser mixed aggregates with pathogens, increasing co-aggregation by 28% (*E. coli*), 17% (*S. aureus*), and 30% (*S. typhi*) (Figure 8D). These data indicate that *S. cerevisiae TAD1-KO* can establish intimate contact with target bacteria, thereby enhancing the local delivery of antimicrobial metabolites within the gut lumen.

### 3.10. S. cerevisiae TAD1-KO Has Antibacterial Therapeutic Effect In Vivo

#### 3.10.1. *S. cerevisiae* TAD1-KO Could Prolong the Survival Time of *G. mellonella*

As an invertebrate, the *G. mellonella* is less sensitive and ethically controversial than mammals [42]. According to the principle of substitution, lower animals can be used as substitutes for higher animals, especially in basic biological or toxicological research [43]. In addition, previous studies have shown that the results of the infection model of *G. mellonella* are similar to those of mammals [44]. An acute toxicity evaluation revealed that single-dose administration of live yeast, heat-killed yeast, or its CFS caused no mortality in *G. mellonella* during the 5-day observation window, confirming an excellent safety profile (Figure 9A). Therapeutic assays showed that both live yeast and its CFS significantly prolonged larval survival after a bacterial challenge, whereas heat-inactivated yeast conferred no protection (Figure 9B,C). The bacterial load results similarly indicate that both CFS and the live yeast group effectively reduced the bacterial load within the larval (Figure 9E,F). These data indicate that the protective effect is mediated by metabolites actively secreted by viable yeast rather than by structural cell components.

#### 3.10.2. *S. cerevisiae* TAD1-KO Has Therapeutic and Protective Effects on Enteritis in Mice Caused by *S. typhi*

*S. typhi* is one of the major pathogens of food-borne diarrhea in the world, and the study of its infection model can directly guide clinical treatment and vaccine development [45]. To evaluate the in vivo efficacy of *S. cerevisiae TAD1-KO* against *S. typhi*, we established a murine infection model. A preliminary 7-day safety study revealed no adverse reactions; *TAD1-KO*- or BY4743-treated mice gained weight in parallel with the 0.9% saline group, confirming low toxicity (Figure 10A). On day 7, animals were challenged with *S. typhi* and monitored for 5 days. *S. cerevisiae TAD1-KO* administration markedly attenuated infection-induced weight loss (Figure 10B) and improved 10-day survival to 66.7% versus 16.7% in 0.9% of the saline controls (Figure 10C; *p* < 0.01), demonstrating a significant protective effect. The spleen is an important immune organ of the organism, and the measurement of colon length is also a key and intuitive pathological evaluation index in the enteritis infection model [46]. Macroscopic and bacteriological analyses corroborated the survival benefit. At day 5 post-challenge, 0.9% saline- and BY4743-treated mice exhibited splenomegaly with congestion, elevated splenic indices, foreshortened and hyperaemic colons, and contracted cecum pouches (Figure 10D,E). In contrast, *TAD1-KO*-treated animals retained normal visceral morphology. Quantitative culture revealed a reduction in *S. typhi* recoverable from the ileum, colon, and caecum (Figure 10F–H; *p* < 0.001). Persistent colonization by *S. cerevisiae TAD1-KO* was confirmed in all intestinal segments (Figure 10I), implying that stable mucosal residence underpins the observed protection. H&E revealed pronounced inflammatory infiltration, epithelial sloughing, and vascular congestion in the ileal sections from 0.9% saline- and BY4743-treated mice (Figure 10J) [47], whereas *TAD1-KO*-treated animals displayed an intact villous architecture, absence of oedema, and negligible leukocyte infiltration, indicating preservation of the intestinal barrier. To further assess the effect of *S. cerevisiae TAD1-KO* on the inflammatory response, we examined the expression levels of key cytokines in ileal tissue. Compared with the 0.9% saline group, *S. cerevisiae TAD1-KO* treatment significantly downregulated pro-inflammatory mediators such as IL-6 and TNF-α, while upregulating the anti-inflammatory IL-10 [48], indicating a shift toward an anti-inflammatory milieu. Microbial infection disrupts the intestinal barrier and increases permeability [46]. We, therefore, quantified tight-junction (TJ) proteins and found that *S. cerevisiae TAD1-KO* markedly enhanced the expression of Occludin, Claudin-1, and ZO-1 relative to the 0.9% saline controls (Figure 10K). Collectively, these data demonstrate that *S. cerevisiae TAD1-KO* mitigates *S. typhi*-induced intestinal injury by dampening inflammation and reinforcing the epithelial barrier.

### 3.11. S. cerevisiae TAD1-KO Inhibits the Proliferation of E. coli, S. aureus, and S. typhi

Cell co-culture is the simultaneous cultivation of two or more different types of cells in the same system, allowing them to interact in a shared environment [49]. This technique is used to simulate the cellular microenvironment in the body and study the interactions between different cells and their effects on biological processes such as signaling, cell growth, and differentiation [50]. Using a direct co-culture model, we assessed whether live *S. cerevisiae TAD1-KO* suppresses pathogen proliferation. After 16 h, viable counts of *E. coli*, *S. aureus*, and *S. typhi* declined markedly, yielding inhibition rates of 89.5%, 55.1%, and 52.2%, respectively (Figure 11A–C). The parental BY4743 strain exerted significantly weaker suppression, underscoring the enhanced antimicrobial competence conferred by *TAD1* deletion.

### 3.12. S. cerevisiae TAD1-KO Produces Antimicrobial Effects by Secreting Active Metabolites

In order to verify whether the secretions of *S. cerevisiae TAD1-KO* are able to inhibit the growth of pathogenic bacteria, we used the Transwell device to test this idea [51]. The transwell system was employed to distinguish contact-dependent from secreted-factor-mediated inhibition. After 16 h, both the mixed and separated culture regimes reduced *E. coli* and *S. aureus* viability by >80% (inhibition rates of 81.7% and 83.7%, respectively; Figure 12A,B), with no statistically significant difference between the two conditions (*p* > 0.05). These data demonstrate that *S. cerevisiae TAD1-KO* exerts its antibacterial activity principally through the release of diffusible metabolites rather than via direct cell–cell contact.

### 3.13. Metabolomics Analyses Suggested That the Metabolites Exerting Antimicrobial Effects in S. cerevisiae TAD1-KO CFS Were Mainly Organic Acids

Metabolomic discrimination was first visualized by unsupervised PCA, which revealed a clear separation between the *TAD1-KO* and BY4743 CFS along PC1/PC2 (Figure 13A), indicating distinct metabolic fingerprints. Supervised PLS-DA reinforced this segregation (Figure 13B) with robust model parameters (R^2^ > 0.99, Q^2^ > 0.95 after 7-fold cross-validation), confirming predictive reliability. HMDB (https://hmdb.ca accessed 14 October 2025) is the largest and most comprehensive organism-specific metabolic database. HMDB annotation showed that organic acids and their derivatives constituted 68.9% of the 177 significantly altered metabolites (Figure 13C). KEGG Compound is a collection of small molecules, biopolymers, and other chemical substances relevant to biological systems. In contrast, KEGG Compound sub-classification ranked amino-acid/peptide derivatives as the second most abundant category (Figure 13D). The clustering heatmap (Figure 13E) illustrated that amino-acid-related metabolites were globally downregulated in *TAD1-KO* CFS, whereas several organic acids—most notably succinic acid—were markedly upregulated. Differential features were rigorously filtered by combining OPLS-DA VIP ≥ 1, |Fold Change| ≥ 1, and *p* < 0.05. The resulting volcano plot (Figure 13F) identified 364 significantly changed metabolites (238 downregulated, 126 upregulated). Among the 126 upregulated entities, the majority were organic acids, including orotic acid, dihydroorotic acid, ureidosuccinic acid, phenylpyruvic acid, and D-3-phenyllactic acid; the VIP score plot (Figure 13G) independently confirmed these hits. Several of these metabolites have documented antibacterial, antioxidant, or immunomodulatory activities, providing a chemical rationale for the enhanced antimicrobial potency of *TAD1-KO* CFS. Succinic acid, also known as butanedioic acid, is a naturally occurring dicarboxylic acid with antimicrobial properties [52]; D-3-phenyllactic acid has been shown to significantly inhibit the growth of a wide range of bacteria and mycobacteria [53]; phenylpyruvic acid is a precursor for the biotransformation of phenyllactic acid, which has been shown to have clear antimicrobial activity [54]. All of the identified acids were elevated > 2.5-fold. KEGG pathway enrichment (hypergeometric test, BH-adjusted *p* < 0.05) revealed significant over-representation of amino-acid metabolism, pyrimidine metabolism, aminoacyl-tRNA biosynthesis, nicotinate/nicotinamide metabolism, and ABC transporters (Figure 13H). Among these, D-amino-acid metabolism was the most prominent [55]; its α-keto-acid products feed directly into the tricarboxylic acid (TCA) cycle and gluconeogenesis, the core hubs for inter-conversion of the upregulated organic acids (citrate, isocitrate, α-ketoglutarate, succinate, fumarate, malate, and oxaloacetate). Thus, *TAD1* deletion redirects carbon flux toward organic acid biosynthesis, thereby augmenting the antimicrobial metabolite pool.

### 3.14. S. cerevisiae TAD1-KO CFS pH Neutralization Attenuates the Inhibitory Effect on E. coli and S. aureus

We measured the pH value of *S. cerevisiae TAD1-KO* CFS to be 4.0 ± 0.37. The results of the pH neutralization experiment in Figure 14 show that when *S. cerevisiae TAD1-KO* CFS is acidic, its inhibitory effect on pathogenic bacteria is significant. However, as *S. cerevisiae TAD1-KO* CFS gradually becomes neutral, the inhibitory effect also deteriorates. Thus, the antimicrobial potency of *S. cerevisiae TAD1-KO* CFS is pH-dependent and attributable, at least in part, to the acidic metabolites identified by our metabolomic survey.

## 4. Discussion

Previous studies have found that probiotics have unique physiological functions. They can treat a variety of diseases, including maintaining the balance of intestinal flora [4], preventing the incidence of tumors and cancers [56], and treating diarrhea [57], as well as prevent and treat high cholesterol, improve lactose intolerance, promote nutrient absorption, and prevent other human diseases [58]. While most preparations are bacterial, probiotic yeasts offer distinct advantages: intrinsic antibiotic resistance permits co-administration with antimicrobial therapy [59], and their larger size provides spatial hindrance against pathogen adhesion [60]. Within the *Saccharomyces* clade, *S. cerevisiae* occurs naturally in plant material and the animal gut, and its cell volume exceeds that of common bacteria several-fold. The related species *S. boulardii* remains the only clinically approved mycobiotic [61], yet it is taxonomically and metabolically distinct from *S. cerevisiae*. Building on our high-throughput knockout screen, we selected *S. cerevisiae TAD1-KO* to dissect its antimicrobial and probiotic attributes. Direct co-culture assays revealed robust, strain-specific suppression of *E. coli*, *S. aureus*, and *S. typhi*, while spectrum tests confirmed broad-spectrum activity. Growth-curve analyses further showed that *S. cerevisiae TAD1-KO* CFS lowers the µmax and stationary-phase density, corroborating its bacteriostatic efficacy—findings that align with Saidi et al. [62], who reported growth inhibition by fruit-derived *S. cerevisiae* CFS but not by cell lysate. Neutralization experiments demonstrated pH-dependence: activity declined sharply as the CFS was titrated toward neutrality, implying that acidic metabolites mediate, or at least potentiate, the antibacterial effect—a scenario analogous to *S. boulardii* acetic acid production at 37 °C [63]. Collectively, these data highlight the therapeutic potential of the acid-active metabolites enriched in *S. cerevisiae TAD1-KO* CFS.

Biofilm formation is a critical survival strategy that enhances environmental persistence and tolerance to both antibiotics and host defenses [64]. The process unfolds in four sequential stages: initial adhesion to biotic or abiotic surfaces, microcolony accumulation, maturation, and dispersal [25]. Probiotic interference at any of these stages can attenuate pathogenicity. For instance, *E. coli* Nissle 1917 has been shown to inhibit biofilms of enteropathogenic *E. coli* strains [65]. Hager et al. [66] used filtrate combined with probiotics to inhibit a pathogenic bacterium–fungus multi-bacterial biofilm. In the present study, *S. cerevisiae TAD1-KO* CFS inhibited biofilm formation by *E. coli*, *S. aureus*, and *S. typhi* at concentrations below the MIC, indicating non-bactericidal antibiofilm activity. Adhesion to host or abiotic surfaces is the prerequisite for biofilm development and infection [67]; consequently, reducing adhesion can prevent disease initiation. However, hydrophobicity is also related to the adhesion and proliferation of the strain in intestinal epithelial cells [68], and high hydrophobicity is conducive to the interaction between probiotics and intestinal epithelial cells [41]. Hydrophobic values ≥ 70% are considered high hydrophobic, between 40% and 70% are moderately hydrophobic, and < 40% are considered low hydrophobic [69]. EPS, composed of polysaccharides, proteins, and nucleic acids, provides the three-dimensional scaffold of biofilms [25]; hence, EPS disruption is essential for biofilm eradication [28]. Quantitative phenotypic assays revealed that *S. cerevisiae TAD1-KO* CFS significantly decreased pathogen adhesion, hydrophobicity, and EPS production in a dose-dependent manner, suggesting that the supernatant compromises early biofilm events. These alterations may limit bacterial docking to intestinal epithelia in vivo, thereby averting infection. To explore the molecular basis, qRT-PCR demonstrated the downregulation of *E. coli* biofilm-associated genes after CFS exposure, mirroring previous reports where *S. cerevisiae* CFS suppressed *icaA* and *sarA* in *S. aureus* [3] and inhibited EPS synthesis and biofilm genes in *Listeria monocytogenes* [70]. Collectively, these data indicate that *S. cerevisiae TAD1-KO* CFS targets both structural and regulatory components of biofilm development, offering a multifaceted strategy for pathogen control.

The FAO and WHO define probiotics as “live microorganisms” which, when administered in adequate amounts, confer a health benefit on the host [71]. Hence, survival throughout the gastrointestinal tract is mandatory. Gastric acid (pH 1.5–3.0) [72] and bile salts (0.03–0.3% in the human small intestine) [73] constitute the first two physiological barriers. In vitro simulation confirmed that *S. cerevisiae TAD1-KO* tolerates both stresses, retaining > 80% viability at pH 3.0 and 0.3% bile. Auto-aggregation ability is also related to the adhesion and proliferation of the strain in intestinal epithelial cells [68]. The auto-aggregation ability of probiotics enables them to reach a relatively high concentration in the gut, which is conducive to adhesion and resistance to harsh environments [41]. An auto-aggregation value of ≥ 60% is considered to have strong auto-aggregation ability, while between 30% and 60% is average, and it is weak when it is lower than 30% [74]. Lara-Hidalgo et al. [75] also observed that *S. boulardii* had a strong auto-aggregation ability. Co-aggregation with pathogens constitutes an additional exclusion mechanism [76]. *S. cerevisiae TAD1-KO* formed denser mixed aggregates with *E. coli*, *S. aureus*, and *S. typhi* than BY4743, which is consistent with the results of previous reports that *S. boulardii* was competitively excluded from *Salmonella* and *Shigella* [77]. Collectively, superior aggregation and co-aggregation capacities enable *S. cerevisiae TAD1-KO* to establish a spatial barrier that limits pathogen adhesion and creates a microenvironment conducive to the delivery of antimicrobial metabolites.

Selection of an appropriate infection model is pivotal for pre-clinical evaluation of antimicrobials. We employed two complementary systems—*G. mellonella* and murine enteritis—to probe the in vivo efficacy of *S. cerevisiae TAD1-KO*. In both models, live yeast or its CFS prolonged survival and elevated survival rates, whereas heat-inactivated cells conferred no protection, paralleling the in vitro co-culture and Transwell data. These concordant observations indicate that metabolic activity, rather than cell biomass, is required for therapeutic benefit, and support the hypothesis that *S. cerevisiae TAD1-KO* generates protective metabolites in situ. *S. typhi*, the aetiologic agent of enteric fever, penetrates the gut via M cells, replicates within intestinal epithelia, and can disseminate haematogenously to deep organs such as spleen and liver [78]. Blocking initial adhesion to enterocytes is, therefore, a critical checkpoint [79]. In the murine model, oral *S. cerevisiae TAD1-KO* not only improved survival but also preserved intestinal architecture, curtailed bacterial translocation, and limited splenic pathology. qRT-PCR revealed the downregulation of pro-inflammatory IL-6 and TNF-α and concurrent upregulation of the anti-inflammatory cytokine IL-10 and tight-junction genes (Occludin, Claudin-1, ZO-1)—findings that are consistent with previous reports [80,81]. Virulence studies confirmed negligible cytotoxicity, underscoring the clinical translational potential of *S. cerevisiae TAD1-KO* as a safe, live biotherapeutic agent.

Robust in vitro and in vivo antibacterial activity, negligible cytotoxicity, high auto-aggregation, superior co-aggregation with pathogens, and demonstrable tolerance to gastric acid and bile salts collectively indicate that *S. cerevisiae TAD1-KO* fulfils the key criteria for a clinically relevant probiotic. We, therefore, proceeded to dissect the underlying antimicrobial mechanisms in detail.

Antioxidant capacity is a valued functional trait, given that ROS contribute to the pathogenesis of arthritis, cardiovascular disorders, and certain cancers [35]. Oxidative stress and inflammation are reciprocally amplifying processes: ROS activate pro-inflammatory signaling cascades, while inflammatory cells release additional ROS and cytokines that exacerbate tissue injury [82]. We, therefore, quantified intracellular ROS in *E. coli* exposed to graded concentrations of *S. cerevisiae TAD1-KO* CFS. Fluorescence intensity rose with increasing CFS concentration, indicating an oxidative burst within the pathogen. However, supplementation with ROS scavengers (VC or thiourea) produced only numerical—yet statistically non-significant—reductions in *E. coli* survival compared with CFS alone, demonstrating that ROS are a secondary byproduct of membrane disruption and do not constitute the primary bactericidal mechanism. The principal antimicrobial effect is attributable to the direct structural damage inflicted by the *S. cerevisiae TAD1-KO* CFS. The integrity and permeability of cell walls and membranes are the basis for bacteria to complete their basic life activities. Cell walls are critical to bacterial morphology and viability, and interfering with the development of disease-causing biofilms in antifungal studies is considered an effective anti-infective strategy [83]. The cell membrane is an important part of cell structure, which can maintain the integrity of cells and has the function of isolating the cell from the outside world. It also plays an important role in receptor recognition, information transmission, and material transport inside and outside the cell [84]. The destruction of the cell membrane will affect cell growth to a certain extent. Consistent with this paradigm, *S. cerevisiae TAD1-KO* CFS increased extracellular AKP and NPN uptake, elevated ONPG hydrolysis, and augmented ROS accumulation, collectively demonstrating compromised wall and membrane integrity (Figure 15). These findings mirror the mechanism described by Huang et al. [85], who reported that cinnamon-leaf-oil constituents (α-terpineol, terpinen-4-ol, and δ-terpineol) dissipate membrane potential, induce AKP leakage, and ultimately kill Gram-negative bacteria through envelope collapse. Thus, *S. cerevisiae TAD1-KO* CFS appears to exert its bactericidal effect via a comparable dual-target attack on the cell wall and membrane.

Previous studies have shown that *S. cerevisiae* secretes antimicrobial substances, including proteases [86], organic acids (such as lactic acid [6]), fatty acids [87], and polyamines [88], to inhibit the growth of pathogenic bacteria. Despite the complex composition of *S. cerevisiae* CFS and metabolites of unknown chemical structure, untargeted metabolomics remains an effective tool for the identification of a large number of metabolites in complex systems [89]. Non-targeted LC-MS/MS metabolomics was employed to dissect the chemical basis of *S. cerevisiae TAD1-KO* CFS activity. Raw datasets (941 positive- and 787 negative-ion features) were filtered, gap-filled, normalized, and log-transformed, yielding 919 and 782 high-quality signals, respectively. Combining univariate t-tests with multivariate OPLS-DA (VIP > 1, *p* < 0.05) identified 364 differential metabolites from a total pool of 2955; 126 were upregulated, predominantly organic acids, whereas 238 were downregulated, mainly amino acids/peptides. Clustering, volcano, and VIP plots independently confirmed organic acids as the principal discriminating class. KEGG enrichment revealed significant over-representation of D-amino-acid metabolism. D-Amino acids are increasingly recognized as bioactive molecules: many impart enhanced sweetness relative to their L-enantiomers during fermentation [55], and nanomolar concentrations of D-Leu, D-Met, D-Tyr, and D-Trp inhibit biofilm formation in *Bacillus subtilis* [90] and *Staphylococcus aureus* [91]. The central enzyme, D-amino-acid oxidase (DAAO) [55], channels D-amino acids into α-keto acids that feed directly into the TCA cycle, linking carbon flux to acidogenesis. In summary, we posit that *TAD1* deletion—encoding tRNA-adenosine deaminase 1—abolishes i^6^A37 tRNA modification [92], thereby attenuating translational elongation kinetics and precipitating a metabolomic decline in free amino acids and low-molecular-weight peptides. Concomitantly, the D-amino-acid catabolic axis is transcriptionally amplified; its deamination-derived α-keto acids anaplerotically recharge the TCA cycle and gluconeogenic core, functioning as the principal carbon-redistribution nodes. By relieving tRNA-modification-dependent translational braking, *TAD1* loss reallocates carbon flux from proteogenesis to the accumulative biosynthesis of short-chain carboxylic acids, thereby expanding the intracellular reservoir of antimicrobial organic acids and their bioactive intermediates. This translational–metabolic rewiring ultimately endows *S*. *cerevisiae* with a gain-of-function bacteriostatic phenotype, establishing a paradigm of “translation–metabolism cross-talk” that potentiates microbial antagonism.

## 5. Conclusions

From a high-throughput screen of 1800 YKOC mutants, we identified *TAD1* as a core genetic determinant of antimicrobial activity. The corresponding knockout strain, *S. cerevisiae TAD1-KO*, displayed potent, broad-spectrum inhibition of pathogenic bacteria, both in vitro and in vivo. Its CFS prevented biofilm formation by decreasing pathogen adhesion, EPS production, and the transcription of biofilm-associated genes. Administration of live *S. cerevisiae TAD1-KO* or its CFS conferred significant protection in *G. mellonella* and murine infection models, while no cytotoxicity was observed in any test system. Mechanistic analyses revealed that high-dose CFS compromises cell–wall and membrane integrity, triggering an intracellular ROS accumulation that underpins the rapid killing observed in time–kill assays. Untargeted metabolomics further demonstrated that these effects are underpinned by a massive over-production of organic acids, particularly those funnelled through D-amino-acid catabolism. Collectively, our data establish *S. cerevisiae TAD1-KO* as a safe, metabolically programmable yeast probiotic that delivers a multi-target antimicrobial arsenal, offering a promising alternative strategy for combating drug-resistant pathogens.

## Figures and Tables

**Figure 1 microorganisms-13-02848-f001:**
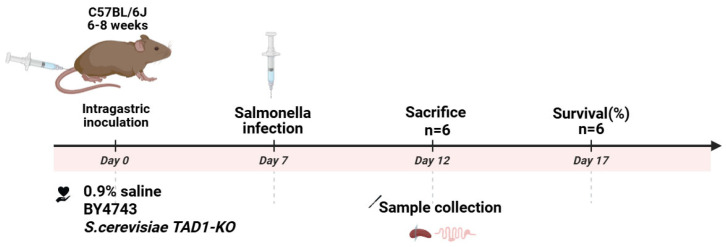
Establishment of a mouse enteritis model.

**Figure 2 microorganisms-13-02848-f002:**
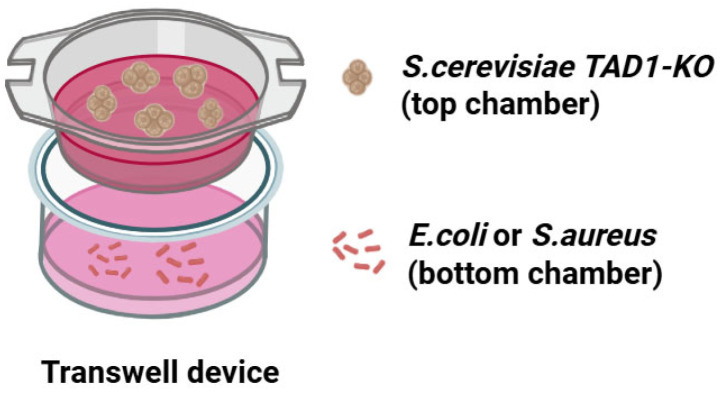
Schematic of the 0.4 µm pore system separating yeast (upper chamber) from bacteria (lower chamber).

**Figure 3 microorganisms-13-02848-f003:**
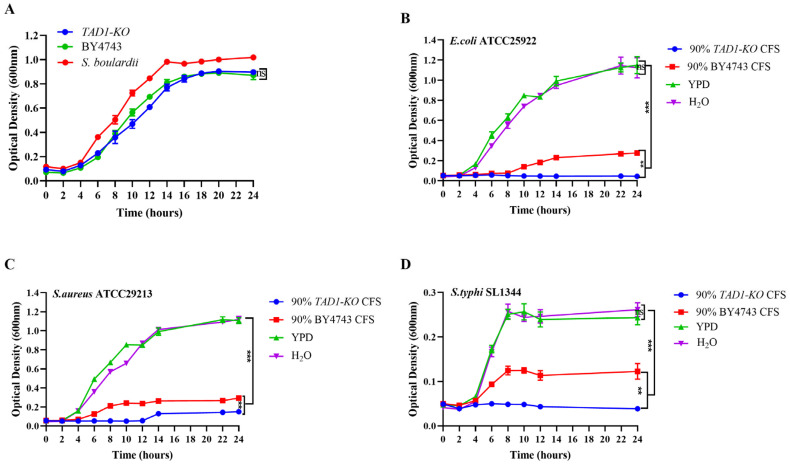
Growth curve measurement. (**A**) Loss of *TAD1* does not impair *S. cerevisiae* growth. Growth curves of parental strain BY4743, *S. boulardii*, and *S. cerevisiae TAD1-KO* cultivated in YPD at 30 °C, 200 rpm for 24 h. The superimposable curves indicate that *TAD1* deletion is growth-neutral under standard laboratory conditions. (**B**–**D**) *S. cerevisiae TAD1-KO* elicits potent metabolite-mediated growth suppression. Growth curves of *E. coli*, *S. aureus*, and *S. typhi* exposed to 90% (*v*/*v*) *S. cerevisiae TAD1-KO*, BY4743, or nutrient controls at 37 °C, 200 rpm. *S. cerevisiae TAD1-KO* CFS significantly delayed lag-to-log transition, reduced µmax, and lowered stationary-phase density. Equivalent growth in sterile water and YPD controls excludes nutrient limitation, confirming inhibition by secreted bioactive metabolites. Data are means ± SD (n = 3); ** *p* < 0.01; *** *p* < 0.001; ns denotes no statistically significant difference.

**Figure 4 microorganisms-13-02848-f004:**
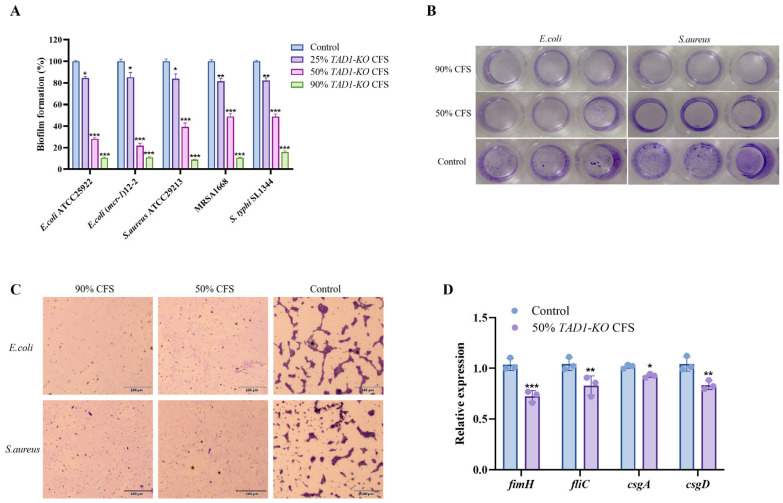
*S. cerevisiae TAD1-KO* CFS inhibits biofilm formation and downregulates biofilm-associated genes: (**A**) Quantitative CV assay showing dose-dependent suppression of biofilms in *E. coli* ATCC25922, *E. coli* (*mcr-1*), *S. aureus* ATCC29213, MRSA, and *S. typhi* SL1344 after 24 h exposure to 25–90% (*v*/*v*) *S. cerevisiae TAD1-KO* CFS. (**B**) 96-well plate images confirming progressive biofilm disruption with increasing CFS concentration. (**C**) Light-microscopy images confirming progressive biofilm disruption with increasing CFS concentration. (**D**) qRT-PCR demonstrating significant downregulation of key biofilm genes (*fimH*, *fliC*, *csgA*, *csgD*) in *E. coli* treated with 50% *S. cerevisiae TAD1-KO* CFS versus untreated control. Data are means ± SD (n = 3); * *p* < 0.05; ** *p* < 0.01; *** *p* < 0.001.

**Figure 5 microorganisms-13-02848-f005:**
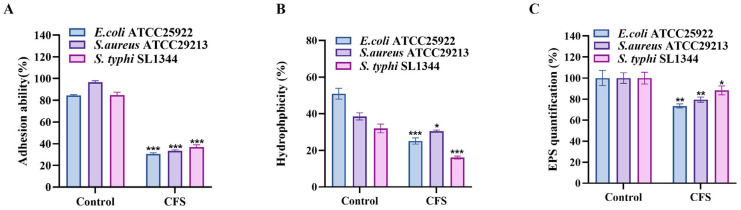
*S. cerevisiae TAD1-KO* CFS impairs adhesion, hydrophobicity, and EPS production: (**A**) Adhesion to glass after 24 h, (**B**) cell-surface hydrophobicity, and (**C**) EPS yield of *E. coli*, *S. aureus*, and *S. typhi* following 24 h exposure to 50% (*v*/*v*) *S. cerevisiae TAD1-KO* CFS. All parameters were significantly reduced versus untreated controls, confirming that CFS targets the early physical determinants of biofilm formation. Data are means ± SD (n = 3); * *p* < 0.05; ** *p* < 0.01; *** *p* < 0.001.

**Figure 6 microorganisms-13-02848-f006:**
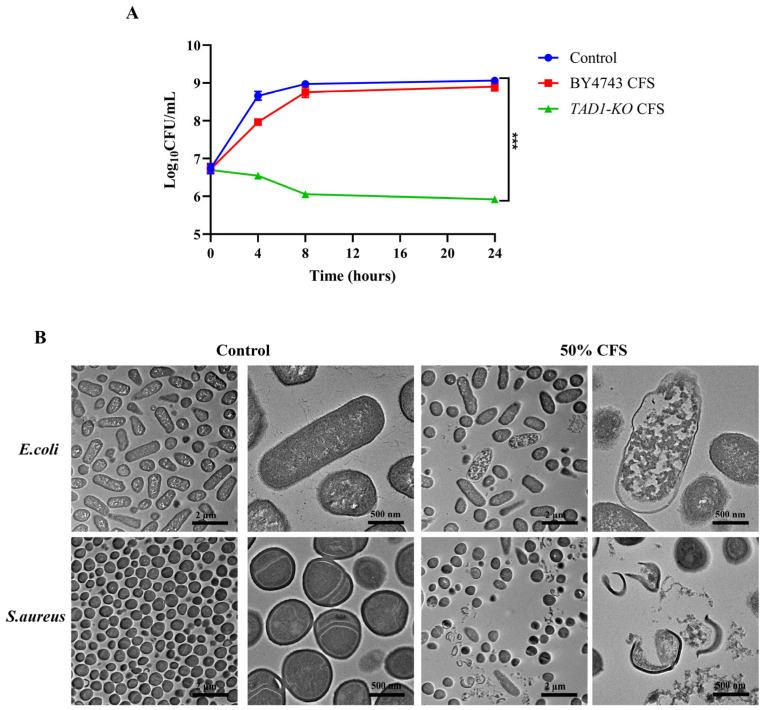
Time–kill kinetics and ultrastructural damage induced by *S. cerevisiae TAD1-KO* CFS. (**A**) Viable counts of *E. coli* exposed to 90% *S. cerevisiae TAD1-KO* CFS, 90% BY4743 CFS, or LB at 37 °C; bactericidal effect could be achieved within 4 h, with maximum bactericidal activity at 8 h. (**B**) Representative TEM micrographs after 6 h treatment with 50% *S. cerevisiae TAD1-KO* CFS: *E. coli* and *S. aureus* exhibit swelling, membrane rupture, cytoplasmic leakage, and plasmolysis compared with intact untreated cells, confirming metabolite-mediated envelope disruption. The black scale in the image represents 2 µm and 500 nm. Data are means ± SD (n = 3); *** *p* < 0.001.

**Figure 7 microorganisms-13-02848-f007:**
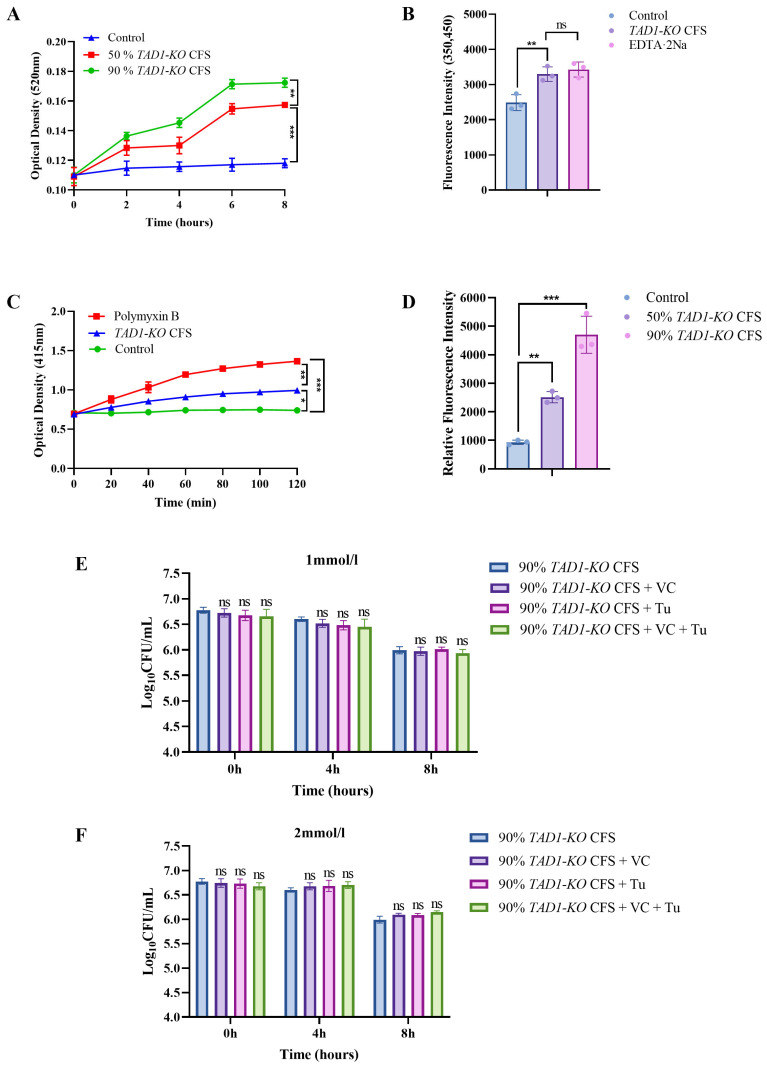
*S. cerevisiae TAD1-KO* CFS sequentially disrupts envelope integrity and elicits intracellular ROS. (**A**) Time-dependent leakage of periplasmic AKP (OD_520_) after 50% *S. cerevisiae TAD1-KO* CFS exposure; significant increase from 4 h onward. (**B**) NPN fluorescence assay: 50% *S. cerevisiae TAD1-KO* CFS raises outer-membrane permeability to the level of 1 mg mL^−1^ EDTA-2Na (positive control). (**C**) ONPG hydrolysis (OD_415_), indicating inner-membrane permeabilization; slightly higher but statistically different from control (* *p* < 0.05). (**D**) DCFH-DA fluorescence: 90% *S. cerevisiae TAD1-KO* CFS triggers robust intracellular ROS accumulation (*** *p* < 0.001), contributing to rapid bacterial killing. (**E**,**F**) ROS scavengers do not impair *S. cerevisiae TAD1-KO* CFS bactericidal efficacy. *E. coli* (1 × 10^7^ CFU mL^−1^) were cultured with 90% *S. cerevisiae TAD1-KO* CFS alone or supplemented with 1 mmol L^−1^ or 2 mmol L^−1^ VC, thiourea, or both. Viable counts (CFU mL^−1^) were determined at 0, 4 and 8 h (37 °C, 200 rpm). No statistically significant difference between groups (*p* > 0.05) indicates that ROS contributes minimally to CFS-mediated killing; the lethal effect is primarily attributable to direct envelope disruption. Data are means ± SD (n = 3); * *p* < 0.05; ** *p* < 0.01; *** *p* < 0.001; ns denotes no statistically significant difference.

**Figure 8 microorganisms-13-02848-f008:**
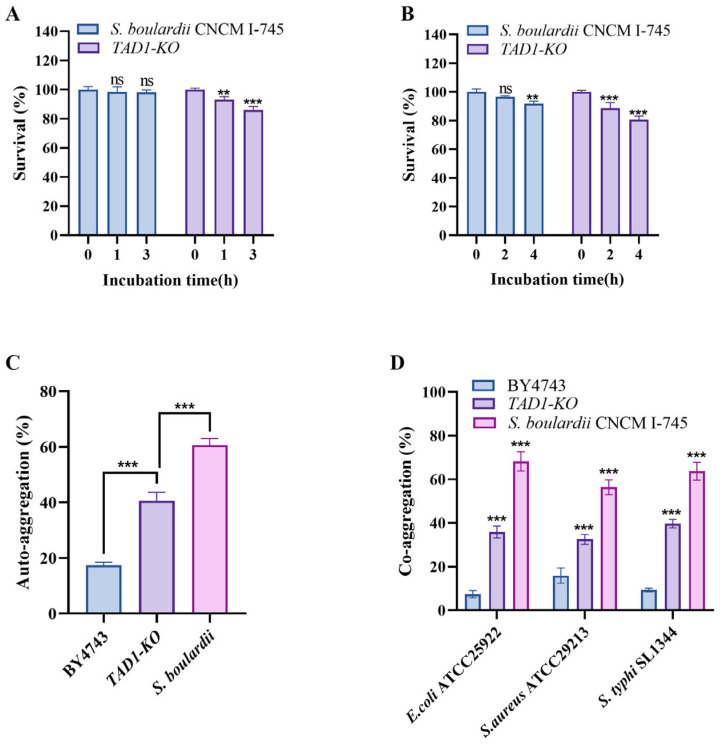
*S. cerevisiae TAD1-KO* tolerates simulated gastrointestinal stress and displays superior aggregation traits. (**A**) Survival after 3 h in simulated gastric fluid (pH 3.0, pepsin 3 mg mL^−1^); *S. cerevisiae TAD1-KO* retained 86.0% viability. (**B**) Survival after 4 h in intestinal fluid (pH 8.0, trypsin 1 mg mL^−1^, 0.3% bile); 80.7% cells remained viable. (**C**) Auto-aggregation at 5 h; *S. cerevisiae TAD1-KO* exceeded BY4743 by 23%. (**D**) Co-aggregation increments with *E. coli*, *S. aureus*, and *S. typhi* (28%, 17%, and 30% higher than BY4743, respectively). Data are means ± SD (n = 3); ** *p* < 0.01; *** *p* < 0.001; ns denotes no statistically significant difference.

**Figure 9 microorganisms-13-02848-f009:**
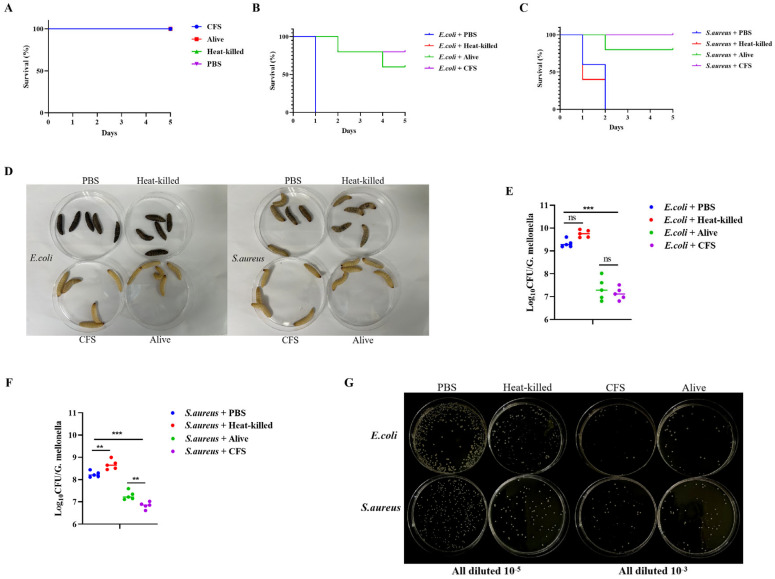
*G. mellonella* toxicity and therapeutic efficacy of *S. cerevisiae TAD1-KO*. (**A**) Survival curves after a single injection of live yeast, heat-killed yeast, or CFS (n = 5 per group); no larval mortality within 5 days confirms low toxicity. (**B**,**C**) Models of infection with *E. coli* and *S. aureus*: live *S. cerevisiae TAD1-KO* and its CFS significantly extended survival (*** *p* < 0.001 vs. PBS), whereas heat-inactivated yeast had no effect, demonstrating that protection requires metabolically active cells. (**D**) Survival rates of *G. mellonella* in each group on the first day of the *E. coli* infection model and the *S. aureus* infection model. (**E**,**F**) On the fifth day post-infection, homogenize *G. mellonella* and calculate the bacterial load within each group. (**G**) After homogenization, each group was spread onto LB agar plates for counting. ** *p* < 0.01; *** *p* < 0.001; ns denotes no statistically significant difference.

**Figure 10 microorganisms-13-02848-f010:**
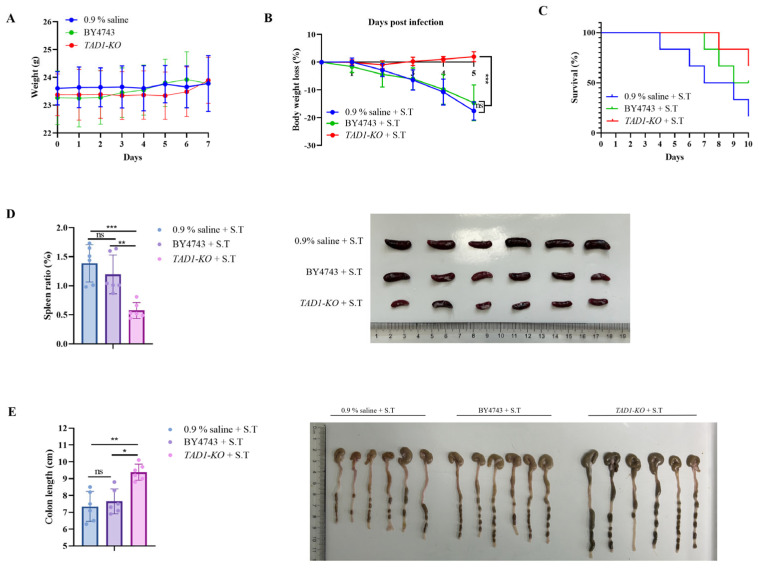
*S. cerevisiae TAD1-KO* protects mice against *S. typhi* infection. (**A**) Seven-day safety: Three groups showed an upward trend in body weight (n = 12). (**B**) Post-infection weight loss attenuated by *S. cerevisiae TAD1-KO* (n = 6). (**C**) Ten-day survival: 66.7% in *S. cerevisiae TAD1-KO* vs. 16.7% in 0.9% saline controls (** *p* < 0.01). (**D**,**E**) Macroscopic pathology at day 5: *S. cerevisiae TAD1-KO* prevents splenomegaly, colon shortening, and cecum congestion. (**F**–**H**) *S. cerevisiae TAD1-KO* reduces the bacterial load of *S. typhi* in the ileum, colon, and caecum (*** *p* < 0.001). (**I**) Stable *S. cerevisiae TAD1-KO* colonization in all gut segments. (**J**) H&E staining of the terminal 2 cm of the ileum sampled from three groups of mice; red arrows indicate inflammatory infiltration, and the scales in the figure are 100 µm and 50 µm, respectively. Intact villi and negligible inflammation in the *S. cerevisiae TAD1-KO* group versus marked infiltration in controls. (**K**) Colon tissue was taken for qRT-PCR assay: downregulation of IL-6, TNF-α, and upregulation of IL-10, Occludin, Claudin-1, and ZO-1 in *S. cerevisiae TAD1-KO* group (n = 6). Data are means ± SD; * *p* < 0.05; ** *p* < 0.01; *** *p* < 0.001; ns denotes no statistically significant difference.

**Figure 11 microorganisms-13-02848-f011:**
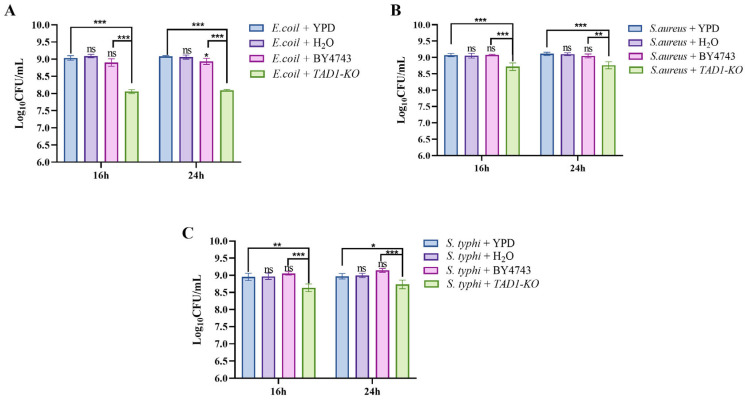
(**A**–**C**) Direct co-culture reveals enhanced pathogen suppression by live *S. cerevisiae TAD1-KO*. Viable counts (CFU mL^−1^) of *E. coli*, *S. aureus*, and *S. typhi* after 16 h contact with live *S. cerevisiae TAD1-KO* or BY4743 at 37 °C, 200 rpm. Inhibition rates: 89.5%, 55.1%, and 52.2% for *S. cerevisiae TAD1-KO* versus ≤ 20% for BY4743, confirming the contribution of *TAD1* deletion to antimicrobial competence. Data are means ± SD; * *p* < 0.05; ** *p* < 0.01; *** *p* < 0.001; ns denotes no statistically significant difference.

**Figure 12 microorganisms-13-02848-f012:**
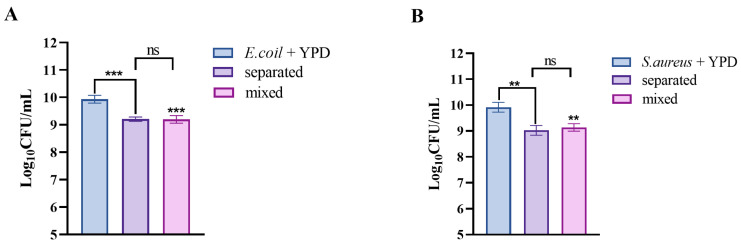
Transwell assay confirms metabolite-mediated antibacterial activity of *S. cerevisiae TAD1-KO*. (**A**,**B**) Viable counts of *E. coli* and *S. aureus* after 16 h at 37 °C. Inhibition rates: mixed culture 81.7%, separated culture 83.7%; no significant difference between regimes (*p* > 0.05, n = 3), demonstrating that diffusible metabolites, not direct contact, drive antibacterial activity. Data are means ± SD; ** *p* < 0.01; *** *p* < 0.001; ns denotes no statistically significant difference.

**Figure 13 microorganisms-13-02848-f013:**
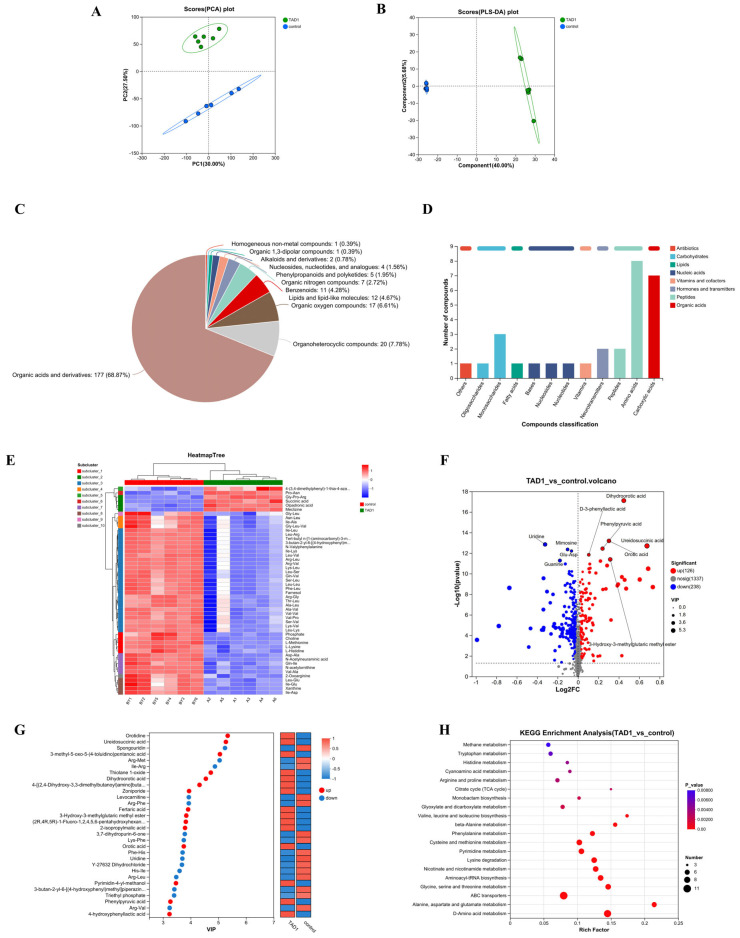
Untargeted metabolomics reveals organic acid enrichment in *S. cerevisiae TAD1-KO*. (**A**) PCA and (**B**) PLS-DA score plots show clear segregation of *S. cerevisiae TAD1-KO* vs. BY4743 CFS (R^2^ > 0.99, Q^2^ > 0.95). (**C**) HMDB: 68.9% of altered metabolites are organic acids. (**D**) KEGG subclass: amino-acid derivatives, second most abundant. (**E**) Clustering heatmap: amino-acid-related metabolites are downregulated, and organic acids are upregulated. (**F**) Volcano plot: 364 differential metabolites (238 down, 126 up); major upregulated acids labeled. (**G**) VIP plot confirms succinic, D-3-phenyllactic, phenylpyruvic, orotic, and dihydroorotic acids among top discriminants (>2.5-fold increase). (**H**) KEGG enrichment: D-amino-acid metabolism is most over-represented (BH-adjusted *p* < 0.05), channelling carbon into TCA cycle organic acids that collectively underpin enhanced antimicrobial activity.

**Figure 14 microorganisms-13-02848-f014:**
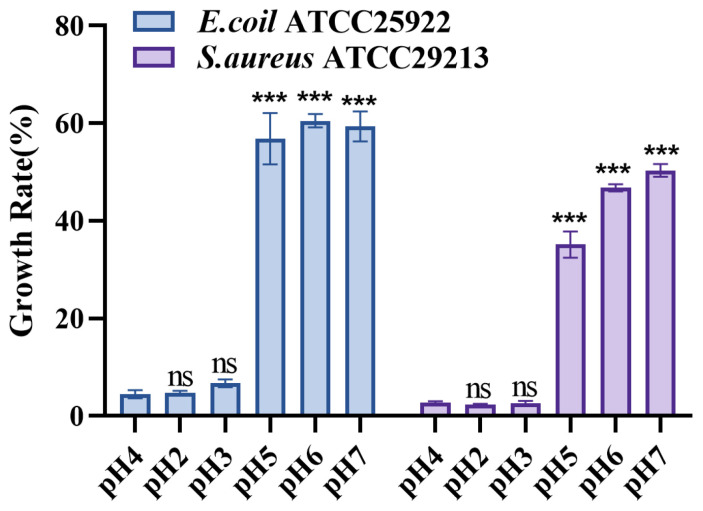
Antimicrobial activity of *S. cerevisiae TAD1-KO* is pH-dependent. Inhibitory efficacy against *E. coli* and *S. aureus* (OD_600_, 24 h) declined sharply as the native CFS (pH 4.0 ± 0.37) was stepwise neutralized to pH 7.0, demonstrating that the bactericidal effect is mediated predominantly by acidic metabolites identified in the metabolomic analysis. Data are means ± SD; *** *p* < 0.001; ns denotes no statistically significant difference.

**Figure 15 microorganisms-13-02848-f015:**
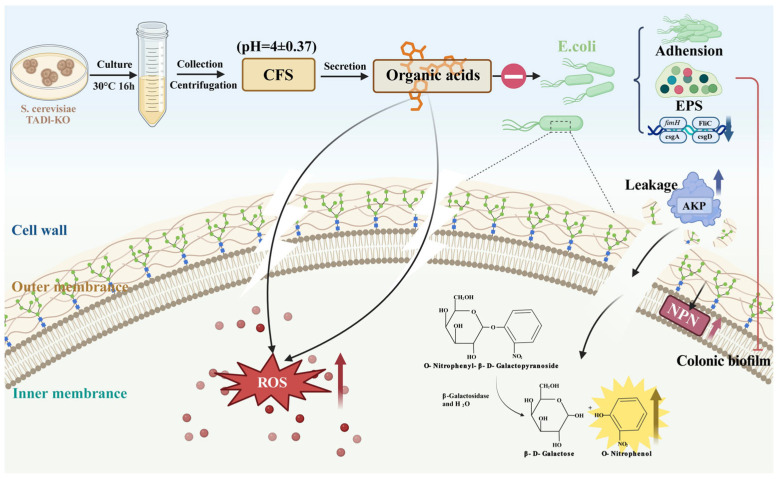
Mechanism diagram of *S. cerevisiae TAD1-KO* against *E. coli*. *S. cerevisiae TAD1-KO CFS* secretes large quantities of organic acids, causing a reduction in the expression of biofilm-associated genes in *E. coli* (blue arrow); CFS disrupts the normal cellular structure of *E. coli*, leading to increased leakage of AKP (blue arrow) and NPN (pink arrow) into the extracellular space, alongside elevated intracellular ONP levels (yellow arrow). Concurrently, organic acids promote intracellular ROS accumulation (red arrow), inducing oxidative stress damage within the cell.

**Table 1 microorganisms-13-02848-t001:** The antibacterial effect of CFS on different bacteria [growth rate (%)].

Strain *	BY4743 CFS	*TAD1-KO* CFS	*S. boulardii* CFS
*E. coli* ATCC25922	50.32% ± 0.93%	11.62% ± 0.64% ***	36.54% ± 0.47% **
*E. coli* (*mcr-1*) 12-2	42.29% ± 0.51%	7.45% ± 0.78% ***	15.24% ± 0.71% **
*S. aureus* ATCC29213	39.9% ± 1.13%	7.27% ± 1.31% ***	40.79% ± 1.1% *
*S. aureus* ATCC25923	37.92% ± 0.87%	5.94% ± 0.98% ***	38.52% ± 0.33% ^ns^
MRSA 1668	42.78% ± 0.31%	8.7% ± 1.17% ***	37.42% ± 0.29% *
*P. aeruginosa* 1554	24.06% ± 0.43%	1.89% ± 0.67% ***	23.44% ± 0.54% ^ns^
*K. pneumoniae* 2118	26.7% ± 0.73%	3.77% ± 1.7% ***	21.31% ± 1.6% *
*A. baumannii* 21-1	21.01% ± 0.51%	1.45% ± 0.59% ***	2.59% ± 0.98% ***
*S. typhi* SL1344	51.33% ± 0.86%	12.55% ± 1.13% ***	31.77% ± 0.58% **

* *E. coli* ATCC25922 is the standard strain of *E. coli*; *Escherichia coli* (*mcr-1*) 12-2 is a clinical isolate of *Escherichia coli*, carrying the polymyxin resistance gene (*mcr-1*); *S. aureus* ATCC29213 and *S. aureus* ATCC25923 are standard strains of *S. aureus*; MRSA 1668 is a clinical isolate of methicillin-resistant *S. aureus*; *P. aeruginosa* 1554 is a clinical isolate of *P. aeruginosa* with resistance; *K. pneumoniae* 2118 is a clinical isolate of *K. pneumoniae* with resistance; *A. baumannii* 21-1 is a clinical isolate of *A. baumannii* with resistance; *S. typhi* SL1344 is a mouse *S. typhi* strain carrying streptomycin resistance. The data are presented as mean ± standard deviation. * *p* < 0.05; ** *p* < 0.01; *** *p* < 0.001; ns denotes no statistically significant difference.

## Data Availability

Data are deposited in Mendeley Data with https://data.mendeley.com/preview/sf53kfgs6r?a=b91a9869-46d0-4400-949a-1dcd1390bddd(accessed 18 November 2025).

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
