# Peer review of "Saccharomyces cerevisiae TAD1 Mutant Strain As Potential New Antimicrobial Agent: Studies on Its Antibacterial Activity and Mechanism of Action"

_microorganisms, 2025, doi:10.3390/microorganisms13122848_

Round 1
Reviewer 1 Report
Comments and Suggestions for Authors
The manuscript proposes that TAD1- Knockout Saccharomyces cerevisiae could be a potential new antimicrobial agent against MDR-resistant bacteria.
The S. cerevisiae knockout strain used in this study was selected by screening among 1800 knockout genomes. The authors identified a mutant deficient in TDA1 gene, whose culture supernatant showed the capability of eradicating multidrug-resistant bacteria of species Escherichia coli, Staphylococcus aureus, Klebsiella pneumoniae, and Salmonella typhi in vitro. To clarify the mode of action of S. cerevisiae TDA1-KO strain, the authors studied many biological aspects of action of CFS (cell-free supernatant) and demonstrated effects such as: proliferation inhibition of different pathogenic bacteria, inhibition of biofilm formation of these bacteria, reduction of adhesion to surfaces, alteration of hydrophobicity, damage to the extracellular and intracellular membrane, and permeability. They also demonstrated that TDA1-KO CFS has good probiotic activity and tolerance to gastric and intestinal fluids. Experiments with G. mellonella showed that S. cerevisiae TDA1-KO has antimicrobial effect in vivo, as it prolonged the survival of this invertebrate, and S. cerevisiae TDA1-KO also has a therapeutic and protective effect in enteritis caused by S. typhi in mice. Using metabolomics analysis, it was suggested that metabolites present in the CSF of S. cerevisiae TDA1-KO, such as organic acids, exert antimicrobial effects.
The methodology is well described and clear, the results are well presented, and the discussion is well conducted. The conclusions are in accordance with the results found, and the references are up-to-date and support the findings.
Minor corrections:
Page 16 (lines 557 to 561), Figure 7 legend, review the text and explain what each graph and figure represents. Some data is missing.
Author Response
Dear reviewer,
Firstly, I am deeply grateful for taking the time to review my manuscript amidst your busy schedule. This marks my first independent piece of writing during my master's studies, and I am immensely appreciative of both your affirmation of the work and the invaluable feedback you have provided. I have carefully considered your comments and revised the manuscript accordingly, highlighting the changes in red throughout the text. I sincerely hope that these revisions and my responses meet with your satisfaction.
Comments 1: Page 16 (lines 557 to 561), Figure 7 legend, review the text and explain what each graph and figure represents. Some data is missing.
Response 1: Firstly, I must apologise for my own carelessness in forgetting to explain D-F. (D) Survival rates of G. mellonella in each group on the first day of E. coli infection model and S. aureus infection model. (E-F) On the fifth day post-infection, homogenise G. mellonella and calculate the bacterial load within each group. (G) After homogenisation, each group was spread onto LB agar plates for counting. Specific corrections may be found in lines 595–597 and 607–610.
Thank you very much for taking the time and trouble to read my reply. Should you notice any further errors, I would be grateful if you could point them out. I look forward to hearing from you.
Sincerely yours,
Yu Zhang
Corresponding author: Ying Li liying@xzhmu.edu.cn Zuobin Zhu: zhuzuobin@xzhmu.edu.cn
Address:209 Tongshan Road, Yunlong District, Xuzhou Medical University, Xuzhou, Jiangsu, China
Reviewer 2 Report
Comments and Suggestions for Authors
The authors analyse the effect of tad1 mutation on the action of S. cerevisiae as an antibacterial agent. The manuscript describes numerous experiments behind the mode of action and is worth publication. However, before that, authors should significantly improve their manuscript.
Major:
- Title should be changed into something like "Saccharomyces cerevisiae tad1 mutant strain as Potential New Antimicrobial Agent: Studies on Its Antibacterial Activity and Mechanism of Action". Use yeast convention to name the gene mutant.
- The introduction is too short. Information on the TAD1 gene and its corresponding protein should be added. Its roles in tRNA modification and translation should be added.
- The screening procedure (if any) using the knockout library and rationale of choosing the tad1 for further experiments should be described. The authors should write the exact phenotype of the BY4743 tad1 strain used in the M&M section and explain why this background, but not that of BY4741/BY4742 was used.
- In the Discussion section, authors should speculate about possible translational effects due to the lack of tRNA modification in tad1 mutant leading to the observed phenotypes.
Minor:
- Include other bacteria tested in keywords.
- Line 43: use full name for S. boulardii. Throughout the manuscript, use full binary name first and then the shortened one for each organism.
- Line 68 and throughout the text: formally, "in vivo", "in vitro" etc. should be written in italics.
- Line 94: growth curves are established, not determined. Growth rate constant/generation time numbers would be useful to show. What is"an enzyme marker " at line 96?
- Line 109: what does "static incubation" mean? No shaking? Why was it not used then?
- Line 123: why write "Overnight log-phase"? I am not sure that E.coli is a log phase."Overnight" is sufficient.
- Line 210: "EPS" should be spelt in full first time. Instead of "Methods 9 Hudrophobicity" write "procedure is similar to the one described in 2.9 section".
- Line 158: rephrase into "Kinetics of E. coli killing by CFS" or so.
- Line 238: rephrase into " Killing model of moth Galleria mellonella".
- In Figure 1 , mention also S. boulardii in 1A in the description below. In 1B-D, water or some buffer/physiological solution was used?
- In Figure 4, yellow bars are not very well seen. Please increase/make clear.
- Line 490: use "Byproduct" instead of "by-Product".
- In Figure 7, the letters are too small and not very well seen. Please increase. same for the Figure 8. Ensure that all labelling is clear at least in the original text. Figure 8 is very big; maybe split into 2 figures?
- Line 559-560: use italics for yeast and bacteria names.
- Line 771: use the short name for the moth investigated.
- References: some text appares in Bold. change to the normal one. Ensure that all organism names are in italics.
Author Response
Dear reviewer,
Firstly, I am deeply grateful for taking the time to review my manuscript amidst your busy schedule. I am immensely appreciative of both your affirmation of the work and the invaluable feedback you have provided. I have carefully considered your comments and revised the manuscript accordingly, highlighting the changes in blue throughout the text. I sincerely hope that these revisions and my responses meet with your satisfaction.
Major:
Comment : Title should be changed into something like "Saccharomyces cerevisiae tad1 mutant strain as Potential New Antimicrobial Agent: Studies on Its Antibacterial Activity and Mechanism of Action". Use yeast convention to name the gene mutant.
Response : I must apologise for making such a basic error. Thank you for pointing it out. I have amended the manuscript title as per your suggestion.
Comment : The introduction is too short. Information on the TAD1 gene and its corresponding protein should be added. Its roles in tRNA modification and translation should be added.
Response : Your comments are most valuable. I have included the following passage in the introduction: In preliminary work, we systematically evaluated the CFS antibacterial activity of 1,800 S. cerevisiae single-gene knockout strains [The Saccharomyces cerevisiae gene-knockout collection (YKOC) ] using Escherichia coli (mcr-1) 12-2 as an indicator strain, with the CFS of parental BY4743 serving as a negative control. This screening yielded 303 mutants exhibiting significant antibacterial activity. Enrichment analysis revealed that the TAD1 gene resides at the core module of the differential gene interaction network. The Saccharomyces Genome Database (SGD) annotates TAD1 as an RNA-specific adenosine deaminase (Tad1p/scADAT1), responsible for catalysing the post-transcriptional modification of tRNAAla at position 37 (adenosine→inosine, A37I), representing an upstream event in tRNA functional regulation. Given that Tad1p-mediated modification occurs at the inception of protein synthesis quality control, its absence may indirectly enhance the synthesis and export of yeast secretory antimicrobial molecules by reshaping the translational landscape, activating non-canonical stress pathways, or reprogramming secondary metabolism. Consequently, a causal link between TAD1 deficiency and the antibacterial phenotype warrants further functional validation and mechanistic elucidation. For details, please refer to lines 74–91 of the manuscript.
Comment : The screening procedure (if any) using the knockout library and rationale of choosing the tad1 for further experiments should be described. The authors should write the exact phenotype of the BY4743 tad1 strain used in the M&M section and explain why this background, but not that of BY4741/BY4742 was used.
Response : I trust you are quite meticulous, but I should like to inform you that the screening procedure was conducted by a former colleague in our research group during her master's studies. She is listed as co-author Liu Cong and assisted me in identifying the TAD1 gene. Her paper is currently undergoing major revisions, so I regret that I am unable to provide you with this protocol and some of the results. However, regarding why TAD1 was selected as the subsequent research subject, I have included the following passage in the introduction section of the paper: In preliminary work, we systematically evaluated the CFS antibacterial activity of 1,800 S. cerevisiae single-gene knockout strains [The Saccharomyces cerevisiae gene-knockout collection (YKOC) ] using Escherichia coli (mcr-1) 12-2 as an indicator strain, with the CFS of parental BY4743 serving as a negative control. This screening yielded 303 mutants exhibiting significant antibacterial activity. Enrichment analysis revealed that the TAD1 gene resides at the core module of the differential gene interaction network. Please refer to lines 74–80 of the manuscript. I have also included information regarding the genotypes of the BY4743 and TAD1 strains in the Materials and Methods sectio: Genotype of BY4743: MATa/α, his3Δ1/his3Δ1, leu2Δ0/leu2Δ0, LYS2/lys2Δ0, met15Δ0/MET15, ura3Δ0/ura3Δ0 ; Genotype of YGL243W (TAD1): :kanMX4. These were obtained from the EUROSCARF collection (Scientific Research and Development GmbH, Oberursel, Germany). Please refer to lines 102–106 of the manuscript. Regarding the selection of BY4743 as the control strain rather than BY4741 or BY4742, I wish to clarify the distinctions between the three: within the YKOC system, BY4743 is the default parental strain for genetic knockout studies conducted in this genetic background. BY4743 is a stable diploid derived from the cross between BY4741 and BY4742, possessing two complete genomes; All three strains carry the identical set of four deficiency markers: leu2Δ0, ura3Δ0, his3Δ1, and met15Δ0. Beyond these differences, the genetic backgrounds of all three strains are derived from S288C, with nearly identical genomic sequences. Their cultivation conditions, morphology, and scientific applications are also largely comparable.
Comment : In the Discussion section, authors should speculate about possible translational effects due to the lack of tRNA modification in tad1 mutant leading to the observed phenotypes.
Response : I have included the following passage in the discussion section: In summary, we posit that TAD1 deletion—encoding tRNA-adenosine deaminase 1—abolishes i⁶A37 tRNA modification [92], thereby attenuating translational elongation kinetics and precipitating a metabolomic decline in free amino acids and low-molecular-weight peptides. Concomitantly, the D-amino-acid catabolic axis is transcriptionally amplified; its deamination-derived α-keto acids anaplerotically recharge the TCA cycle and gluconeogenic core, functioning as the principal carbon-redistribution nodes. By relieving tRNA-modification-dependent translational braking, TAD1 loss reallocates carbon flux from proteogenesis to the accumulative biosynthesis of short-chain carboxylic acids, thereby expanding the intracellular reservoir of antimicrobial organic acids and their bioactive intermediates. This translational–metabolic rewiring ultimately endows Saccharomyces cerevisiae with a gain-of-function bacteriostatic phenotype, establishing a paradigm of “translation–metabolism cross-talk” that potentiates microbial antagonism. Please refer to lines 905–918 of the manuscript.
Minor:
Comment : Include other bacteria tested in keywords.
Response : Thank you very much for your suggestion. I have now included Staphylococcus aureus and Salmonella typhi in the keywords, as shown on line 32 of the manuscript.
Comment : Line 43: use full name for S. boulardii. Throughout the manuscript, use full binary name first and then the shortened one for each organism.
Response : Thank you for your kind reminder. I have now supplemented this as S. boulardii CNCM I-745 and spelled out the full name of the abbreviation upon its first appearance throughout the manuscript, with subsequent instances using the abbreviation as shown at line 43 of the manuscript.
Comment : Line 68 and throughout the text: formally, "in vivo", "in vitro" etc. should be written in italics.
Response : First, I must apologise for my own carelessness. I have now italicised all instances of 'in vivo', 'in vitro', 'in suit', and the names of certain bacteria, fungi, and organisms throughout the entire manuscript, as shown in the newly uploaded manuscript.
Comment : Line 94: growth curves are established, not determined. Growth rate constant/generation time numbers would be useful to show. What is"an enzyme marker " at line 96?
Response : Thank you for raising this point. I believe my manuscript requires further refinement in terms of linguistic expression. I have changed it to "Establish Growth Curves of Three Yeasts" as per your suggestion; furthermore, I have provided more detailed specifications for the cultivation conditions of the three yeast strains: 30 °C, 200 rpm for 24 hours; absorbance at 600 nm was measured using a multi-mode microplate reader. Please refer to lines 116–119 of the manuscript.
Comment : Line 109: what does "static incubation" mean? No shaking? Why was it not used then?
Response : I apologise for my linguistic inaccuracies. Not static incubation, I have changed it to "After incubation at 37 °C, 90 rpm for 18-24 h, OD₆₀₀ was read." Please refer to lines 131 of the manuscript.
Comment : Line 123: why write "Overnight log-phase"? I am not sure that E.coli is a log phase."Overnight" is sufficient.
Response : I should imagine you are a meticulous scholar. At your suggestion, I have changed "Overnight log-phase" to "Overnight" in other parts of the manuscript. Please refer to lines 145 of the manuscript.
Comment : Line 210: "EPS" should be spelt in full first time. Instead of "Methods 9 Hudrophobicity" write "procedure is similar to the one described in 2.9 section".
Response : I have spelt out all abbreviations appearing for the first time in the manuscript, such as: Exopolysaccharides (EPS); I have changed "Methods 9 Hudrophobicity" to “Procedure is similar to the one described in 2.9 section.” Please refer to lines 175-176 of the manuscript.
Comment : Line 158: rephrase into "Kinetics of E. coli killing by CFS" or so.
Response : The title you gave was so precise that I have changed it to "Kinetics of E. coli killing by CFS". Please refer to lines 182 of the manuscript.
Comment : Line 238: rephrase into " Killing model of moth Galleria mellonella".
Response : As above, I have also changed it to "Killing model of moth Galleria mellonella". Please refer to lines 264 of the manuscript.
Comment : In Figure 1 , mention also S. boulardii in 1A in the description below. In 1B-D, water or some buffer/physiological solution was used?
Response : I'm terribly careless. I have changed to "Growth curves of parental strain BY4743, S. boulardii and S. cerevisiae TAD1-KO cultivated in YPD at 30 °C 200 rpm for 24 h. " Please refer to lines 375 of the manuscript. Regarding Figures B-D, we employed sterile water as the control. Please refer to lines 381 of the manuscript.
Comment : In Figure 4, yellow bars are not very well seen. Please increase/make clear.
Response : Following your advice, I have not only enlarged the scale bar on the transmission electron micrograph but also increased the resolution and size of the images in the manuscript before re-uploading them.
Comment : Line 490: use "Byproduct" instead of "by-Product".
Response : I have changed to "Byproduct", please refer to lines 533 of the manuscript.
Comment : In Figure 7, the letters are too small and not very well seen. Please increase. same for the Figure 8. Ensure that all labelling is clear at least in the original text. Figure 8 is very big; maybe split into 2 figures?
Response : As above, I have inserted the mouse experimental workflow diagram and chamber apparatus diagram into the Materials and Methods section. Figure 8 has been split into two images, and the remaining figures have been amended and re-uploaded.
Comment : Line 559-560: use italics for yeast and bacteria names.
Response : I apologise for my lack of expertise; I have italicised all bacterial and yeast names throughout the manuscript.
Comment : Line 771: use the short name for the moth investigated.
Response : yes,I have changed to “G. mellonella”.
Comment : References: some text appares in Bold. change to the normal one. Ensure that all organism names are in italics.
Response : I have adjusted the reference font to normal, reformatted in-text references to match the main text rather than using superscript, and italicised all organism names.
Finally, I should like to express my gratitude once more for taking the time and effort to read my article. As my first independently authored piece during my master's studies, your affirmation undoubtedly serves as a driving force for my future endeavours. I am certain there remain numerous imperfections, and I would be most grateful if you could point them out. I shall endeavour to make the necessary amendments and look forward to your response.
Sincerely yours,
Yu Zhang
Corresponding author: Ying Li liying@xzhmu.edu.cn Zuobin Zhu: zhuzuobin@xzhmu.edu.cn
Address:209 Tongshan Road, Yunlong District, Xuzhou Medical University, Xuzhou, Jiangsu, China
Reviewer 3 Report
Comments and Suggestions for Authors
Dear Authors,
I have read your manuscript and found it very interesting. It is extensive research with many results. However, to improve the quality of your manuscript, some corrections are necessary. You can find my comments in the PDF file.
Greetings!

Author Response
Dear reviewer,
Firstly, I am deeply grateful for taking the time to review my manuscript amidst your busy schedule. I am immensely appreciative of both your affirmation of the work and the invaluable feedback you have provided. I have carefully considered your comments and revised the manuscript accordingly, highlighting the changes in green throughout the text. I sincerely hope that these revisions and my responses meet with your satisfaction.
General comment: References should be written throughout the entire paper in line with the text,
not in superscript.
Response : Firstly, I must apologise for my own carelessness. I have amended the citation formats within the manuscript to align with the main text, rather than using superscripts. Please refer to the newly uploaded manuscript.
Authors: Author list and affiliations should be clearly written. Please add the initials of each author.
Response : I have amended the abbreviated names of authors in the final author contributions section to their full names, as detailed in lines 938–942.
Keywords: line 31 - please add space before TAD1
Response : Thank you very much for your meticulous attention. I have now inserted the space; please refer to line 31 for the specific change.
Introduction:
Line 68, line 72-73 – in vitro should be written in italic form
Line 75- in vivo in italic form
Response : Firstly, I must apologise for my own carelessness. I have already amended numerous in vivo, in vitro, in situ, and other undiscovered bacterial and yeast names in the manuscript to italic format.
Materials and methods:
Line 152 – add number 2 to methods 9. (= 2.9.)
Lines 151, 177, 185, 191, 199, 205 – please explain what the abbreviations means: EPS, AKP, NPN,
ONPG, DcFH-DA, VC
Line 304 – remove space between µ and L
Line 316 – please add method description of LC-MS/MS analysis of metabolites (add to
Supplementary)
Line 325 – please add information about GraphPad Prism software used (producer, country)
Response : I am most grateful for these questions you have raised, as they are precisely points I had not considered. Firstly, I have already amended line 176 to 2.9 section; I neglected to spell out the acronym in full upon its first appearance in the manuscript, I have now amended this to: Exopolysaccharides (EPS), alkaline phosphatase (AKP), N-Phenyl-1-naphthylamine (NPN), O-Nitrophenyl-β-d-galactopyranoside (ONPG), Reactive Oxygen Species (ROS), Vitamin C (VC), Thiourea (Tu), please refer to lines 175, 201, 209, 217, 220 and 233-235 of the manuscript; I have also removed space between µ and L, see lines 334-338; I have also uploaded the LC-MS/MS analytical method description for metabolites and all relevant documentation to the supplementary files; for experimental data analysis, I selected GraphPad Prism 10 software, an analytical programme developed in the United States, see lines 361.
Results
General comment for Figures: Figure captions 1–12 are too long; they should only describe what the
figures represent, without including comments on results. Comments of the results should be
included in the main text.
Line 541, line 559-560 – Latin names of the organism and microorganisms should be in italic form
Line 554 – Figure 7- what represent Figures 7D-G? Please add to figure caption.
Line 563 – in vivo should be in italic form
Line 564 and line 597 – remove Figure 8A (Experimental timeline) to Material and methods section
Line 638 - remove Figure 10A to Material and methods section
Line 649, line 651 - please explain what the abbreviations means: HMDB, KEGG
Line 658-659 – write compound names with lowercase initial letters
Response : I am certain my results contain numerous imperfections, and I am most grateful for your assistance in identifying them. Firstly, you are quite right that my figure captions were excessively lengthy, I have now shortened them as much as possible, please refer to the captions beneath the figures in the manuscript. Secondly, the names of bacteria, yeasts, and other organisms mentioned in the text have been changed to italics, please see lines 584-591 in the manuscript. I must apologise for my oversight in failing to explain parts D-G of Figure 9. "(D) Survival rates of G. mellonella in each group on the first day of E. coli infection model and S. aureus infection model; (E-F) On the fifth day post-infection, homogenise G. mellonella and calculate the bacterial load within each group; (G) After homogenisation, each group was spread onto LB agar plates for counting"; see lines 606–609 of the manuscript. I have reinserted the mouse experimental flowchart and chamber apparatus diagram into the Methods section as per your suggestion, and removed the two figures from the Results section. "HMDB (https://hmdb.ca) is the largest and most comprehensive organism-specific metabolic database,KEGG Compound is a collection of small molecules, biopolymers, and other chemical substances relevant to biological systems", I have incorporated the explanations of the two terms into lines 702–708 of the manuscript; "orotic acid, dihydroorotic acid, ureidosuccinic acid, phenylpyruvic acid, D-3-phenyllactic acid", the initial letters of these compound names have also been lowercased, as seen in lines 714–715 of the manuscript.
Discussion
Lines 745, 772, 774, 777, 788, 850 – in vitro and in vivo, in situ should be in italic form
Response : I am truly grateful that you have examined it so thoroughly, and I apologise once again for my carelessness. I have already amended numerous in vivo, in vitro, in situ, and other undiscovered bacterial and yeast names in the manuscript to italic format.
Author Contributions
Please in the list of authors on page 1, indicate initials and align them accordingly here.
Response : I must apologise for not clearly identifying the authors' names. I did not add abbreviations to the authors' names on the first page as you requested, but I have completed the names in the final authors' contributions section. Please refer to lines 937–941 of the manuscript for specifics.
Finally, I should like to express my gratitude once more for taking the time and effort to read my article. As my first independently authored piece during my master's studies, your affirmation undoubtedly serves as a driving force for my future endeavours. I am certain there remain numerous imperfections, and I would be most grateful if you could point them out. I shall endeavour to make the necessary amendments and look forward to your response.
Sincerely yours,
Yu Zhang
Corresponding author: Ying Li liying@xzhmu.edu.cn Zuobin Zhu: zhuzuobin@xzhmu.edu.cn
Address:209 Tongshan Road, Yunlong District, Xuzhou Medical University, Xuzhou, Jiangsu, China
Thank you very much for taking the time and trouble to read my reply. Should you notice any further errors, I would be grateful if you could point them out. I look forward to hearing from you.
Sincerely yours,
Yu Zhang
Corresponding author: Ying Li liying@xzhmu.edu.cn Zuobin Zhu: zhuzuobin@xzhmu.edu.cn
Address:209 Tongshan Road, Yunlong District, Xuzhou Medical University, Xuzhou, Jiangsu, China
Reviewer 4 Report
Comments and Suggestions for Authors
The aim of the presented study was to assess potential probiotic utility of the S. cerevisiae mutant lacking the TAD1 gene. The research is truly wide-ranging and utilises many different biological and molecular methods. The description of the methods seems sufficient, but most of them lack references to the literature, explanations of some of the acronyms used or the names of commercial tests in several cases. Overall, the study is fairly well designed and the results are sufficiently documented. The subject of the research is topical and will certainly attract attention.
The authors first demonstrated that deletion of the TAD1 gene in the S. cerevisiae strain does not significantly affect its ability to multiply, and that culture supernatant (CFS) inhibits the proliferation and biofilm formation of many pathogenic bacteria. Using qRT-PCR, significant downregulation of key E. coli biofilm genes exposed to CFS was demonstrated. A further series of experiments showed that TAD1-KO CFS can reduce adhesion, surface hydrophobicity and EPS production by E. coli, S. aureus and S. typhi. Killing time analysis and TEM imaging revealed the bactericidal activity of TAD1-KO CFS against E. coli and its ability to damage the integrity of the E. coli cell wall and cell membrane permeability. Subsequently, it was demonstrated that the bactericidal activity of TAD1-KO CFS results primarily from direct structural damage to the cell membrane and cell wall, and secondarily from induced intracellular ROS. Finally, metabolomic analyses suggested that the antibacterial activity of S. cerevisiae TAD1-KO CFS is associated with organic acids production and diminish after neutralisation.
Minor:
Page 2/45,56 - references 4, 5, and 9 seem to be inappropriate, please check.
Please add the references referring the methods used, especially adhesion assay (2.8.), hydrophobicity assay (2.9), EPS production (2.10), etc.
AKP, NPN, ONPG, VC - please state the full name when first used, as well as the name of the commercial test/manufacturer.
Does Table 1 show survival rates or the percentage of cells killed?
Author Response
Dear reviewer,
Firstly, I am deeply grateful for taking the time to review my manuscript amidst your busy schedule. I am immensely appreciative of both your affirmation of the work and the invaluable feedback you have provided. I have carefully considered your comments and revised the manuscript accordingly, highlighting the changes in orange throughout the text. I sincerely hope that these revisions and my responses meet with your satisfaction.
Comment:Page 2/45,56 - references 4, 5, and 9 seem to be inappropriate, please check.
Response:I should imagine you are a most meticulous scholar. In response to your suggestion, I have replaced the three references here. [4]Kaźmierczak-Siedlecka K, Ruszkowski J, Fic M, Folwarski M, Makarewicz W. Saccharomyces boulardii CNCM I-745: A Non-bacterial Microorganism Used as Probiotic Agent in Supporting Treatment of Selected Diseases. Curr Microbiol. 2020, 77(9):1987-1996. [5]Stier H, Bischoff SC. Influence of Saccharomyces boulardii CNCM I-745on the gut-associated immune system. Clin Exp Gastroenterol. 2016, 9:269-279. [9]Shen Y, Bai X, Wang J, Zhou X, Meng R, Guo N. Inhibitory Effect of Non-Saccharomyces Starmerella bacillaris CC-PT4 Isolated from Grape on MRSA Growth and Biofilm. Probiotics Antimicrob Proteins. 2025, 17(1):227-239. And rephrased the sentence preceding reference 9: ”For example, it was found that S. cerevisiae cell-free supernatant (CFS) had a minimum inhibitory concentration (MIC) as low as 8% against methicillin-resistant Staphylococcus aureus (MRSA) and was able to inhibit biofilm formation and downregulate the expression of related genes [9].“ Please refer to lines 43–46 and 55–59 in the manuscript.
Comment:Please add the references referring the methods used, especially adhesion assay (2.8.), hydrophobicity assay (2.9), EPS production (2.10), etc.
Response:In the Materials and Methods section, I have not added any references. Methodological citations can be found at the beginning of each corresponding results paragraph. For example: experimental procedures related to adhesiveness, hydrophobicity, and EPS production can be found in the four references inserted at lines 440–445 of the manuscript.
Comment:AKP, NPN, ONPG, VC - please state the full name when first used, as well as the name of the commercial test/manufacturer.
Response:I apologise for my carelessness. I have spelt out all abbreviations in full within the manuscript, such as:"alkaline phosphatase (AKP) ,N-Phenyl-1-naphthylamine (NPN),O-Nitrophenyl-β-d-galactopyranoside (ONPG),Vitamin C (VC),Thiourea (Tu)". Please refer to lines 202, 209, 217, and 234 of the manuscript. The AKP and ROS kits were both purchased from Beyotime, Shanghai; while other powders were acquired from Sangon, Shanghai. Please refer to lines 202 and 225 of the manuscript.
Comment:Does Table 1 show survival rates or the percentage of cells killed?
Response:I apologise for not having clearly indicated this in the table. I have added the following after the table heading:" [growth rate (%)]”.Please refer to line 403 of the manuscript.
Finally, thank you for dedicating your precious time and attention to reviewing my work. It is my honour! I look forward to your reply.
Sincerely yours,
Yu Zhang
Corresponding author: Ying Li liying@xzhmu.edu.cn Zuobin Zhu: zhuzuobin@xzhmu.edu.cn
Address:209 Tongshan Road, Yunlong District, Xuzhou Medical University, Xuzhou, Jiangsu, China
Round 2
Reviewer 2 Report
Comments and Suggestions for Authors
I appreciate the responses and efforts made by the authors to improve the manuscript. However, still some corrections have to be made before paper could be considered for publication.
- Title: if you use capital letters in the title, use all of them: "Saccharomyces cerevisiae tad1 Mutant Strain as Potential New Antimicrobial Agent: Studies on Its Antibacterial Activity and Mechanism of Action".
- Abstract, line 20: use short name for bacteria (like E. coli) since full names were already mentioned above. Also in "in vitro" both words should be in italics.
- Abstract, line 21: CFS and ROS has to be spell out in full when used for the first time.
- Line 43: write S. boulardii in full.
- Line 45; delete "(Replace two references)."
- Line 46: shorten to "S. cerevisiae."
- Line 50: remove "proteins".
- Line 57: remove "(Replace reference)".
- Line 81: insert reference to SGD.
- Line 97: write city, state, country for Invitrogen .
- Line 99; same for French Encyclopedia Pharmaceutical Company. Is it the correct name?
- Line 103: shorten names to E. coli etc.
- Line 112: change title to "Establishing yeast growth curves".
- Line 141: change "was" to "were".
- Line 196: how exactly AKP activity was measured? Should it be "...activity at 520 nm by measuring absorption of phenylphosphate degradation products"?
- Line 197: use "recommendations" instead of "formula".
- Line 212: write "...absorbance of resulting O-nitrophenol at 415 nm".
- Line 223: specify vitamin C (VC) first here, not in line 226.
- Line 308: use capital letters only for the first word. Remove "(moved up from the results section)".
- Line 334: remove "(moved up from the results section)".
- Line 415 : remove the text.
- Line 426: write for the first time "EPS" in full.
- Line 430: write "feature" instead of "hydrophobicity".
- Line 460, figure 6. The size bars care still now very well seen. Place those below the picture and covert the color to black.
- Line 517: write "As" in capital.
- Line 527: write "thiourea" in small letter.
- Lines 808 and 810: no need for hyphen in "in-vitro".
- Line 824: same as above.
- Line 857: Figure 15: remove "ONPG" and "ONP" from the figure.
- Line 888: write "S. cerevisiae".
- References: write organism names in italics.
- Line 1126: remove the additional text.
Author Response
Dear reviewer:
Firstly, I am delighted to receive your correspondence once more. We are grateful to the reviewers for their thoughtful scrutiny and constructive feedback. We concur with your suggestions and have incorporated them into the manuscript, with all amendments highlighted in blue type.
Comments: Title: if you use capital letters in the title, use all of them: "Saccharomyces cerevisiae tad1 Mutant Strain as Potential New Antimicrobial Agent: Studies on Its Antibacterial Activity and Mechanism of Action".
Response: I apologise for my oversight; I have now capitalised all initial letters in the title within the manuscript, as per lines 2–4.
Comments: Abstract, line 20: use short name for bacteria (like E. coli) since full names were already mentioned above. Also in "in vitro" both words should be in italics.
Response: Thank you for your correction. I have amended it to "E. coli, S. aureus, K. pneumoniae and S. typhi in vitro", as per line 20.
Comments: Abstract, line 21: CFS and ROS has to be spell out in full when used for the first time.
Response: I have amended it to “cell-free supernatant (CFS) , reactive oxygen species (ROS)” , as per line 22.
Comments: Line 43: write S. boulardii in full.
Response: I have amended it to“Saccharomyces boulardii CNCM I-745”, as per line 43.
Comments: Line 45; delete "(Replace two references)."
Response: Deleted.
Comments: Line 46: shorten to "S. cerevisiae."
Response: I have amended it to“S. cerevisiae”.
Comments: Line 50: remove "proteins".
Response: Deleted.
Comments: Line 57: remove "(Replace reference)".
Response: Deleted.
Comments: Line 81: insert reference to SGD.
Response: Following your suggestion, I have inserted the URL (https://www.yeastgenome.org/) for the SGD website, as per line 81. Searching directly for the TAD1 gene name on this site will display the corresponding functional information.
Comments: Line 97: write city, state, country for Invitrogen .
Response: I have amended it to “Invitrogen (Carlsbad, California, United States)”, as per line 101.
Comments: Line 99; same for French Encyclopedia Pharmaceutical Company. Is it the correct name?
Response: I have amended it to “Biocodex (France)”, as per line 103.
Comments: Line 103: shorten names to E. coli etc.
Response: I have amended it to “E. coli, S. aureus, P. aeruginosa, K. pneumoniae, A. baumannii and S. typhi”, as per line 107.
Comments: Line 112: change title to "Establishing yeast growth curves".
Response: I have amended it to "Establishing yeast growth curves", as per line 116.
Comments: Line 141: change "was" to "were".
Response: Amended, see line 145.
Comments: Line 196: how exactly AKP activity was measured? Should it be "...activity at 520 nm by measuring absorption of phenylphosphate degradation products"?
Response: Thank you very much for your advice; it was my own fault for not expressing myself clearly. I have amended it to “Cells were centrifuged (10000 × g, 5 min), and the absorbance of the bacterial solution at 520 nm was measured using the AKP Assay kit (Beyotime, Shanghai). ” Regarding the specific detection procedure, I followed the steps provided in the AKP kit manufactured by Beyotime Company. See line 202 for details.
Comments: Line 197: use "recommendations" instead of "formula".
Response: Amended, see line 204.
Comments: Line 212: write "...absorbance of resulting O-nitrophenol at 415 nm".
Response: Thank you for your suggestions, I have amended it to “ONPG hydrolysis was monitored every 20 min by measuring absorbance of resulting O-nitrophenol at 415 nm.” Please see line 220-221.
Comments: Line 223: specify vitamin C (VC) first here, not in line 226.
Response: I have amended the first instances of VC and Tu to read "Vitamin C (VC) or thiourea (Tu)", with all subsequent occurrences abbreviated. Please refer to line 234.
Comments: Line 308: use capital letters only for the first word. Remove "(moved up from the results section)".
Response: I have amended it to “Establishment of a mouse enteritis model.” Please refer to line 320.
Comments: Line 334: remove "(moved up from the results section)".
Response: Deleted.
Comments: Line 415 : remove the text.
Response: Deleted. And position the text and images appropriately. Please refer to line 426-427.
Comments: Line 426: write for the first time "EPS" in full.
Response: The full name of EPS has already been written in full in the Methods and Materials section above, hence it is not written in full here.
Comments: Line 430: write "feature" instead of "hydrophobicity".
Response: Thank you for your suggestion and the changes have been made. Please refer to line 442.
Comments: Line 460, figure 6. The size bars care still now very well seen. Place those below the picture and covert the color to black.
Response: I sincerely apologise that the previous modifications did not meet with your satisfaction. I have redrawn the black scale as per your suggestion and positioned it in the bottom-right corner of the image. Furthermore, I have added the following to the figure caption: “The black scale in the image represents 2 µm and 500 nm.” Please refer to line 478-479.
Comments: Line 517: write "As" in capital.
Response: I have amended it to“ROS As a Byproduct”. Please refer to line 532.
Comments: Line 527: write "thiourea" in small letter.
Response: I have amended it to "VC and thiourea". Please refer to line 542.
Comments: Lines 808 and 810: no need for hyphen in "in-vitro".
Response: I have amended it to "in vitro, in vitro". Please refer to line 832 and 834.
Comments: Line 824: same as above.
Response: I have amended it to "in vitro, in vitro". Please refer to line 848.
Comments: Line 857: Figure 15: remove "ONPG" and "ONP" from the figure.
Response: It has been removed from the image and re-uploaded to Figure. Please refer to line 882.
Comments: Line 888: write "S. cerevisiae".
Response: I have amended it to"S. cerevisiae". Please refer to line 914.
Comments: References: write organism names in italics.
Response: I must apologise for failing to make the changes when you last reminded me. I have now italicised all organism names in the references and highlighted the text in blue.
Comments: Line 1126: remove the additional text.
Response: Deleted. Please refer to line 1153.
Thank you once again for your time and effort. Should you have any further queries, please do not hesitate to inform me. I shall endeavour to present readers with a more polished manuscript. I look forward to your response.
Yours sincerely,
Yu Zhang